# CLN3 deficiency leads to neurological and metabolic perturbations during early development

Ursula Heins-Marroquin[1] , Randolph R Singh[1,2] , Simon Perathoner[3] , Floriane Gavotto[1] , Carla Merino Ruiz[4,5],
Myrto Patraskaki[1], Gemma Gomez-Giro[1], Felix Kleine Borgmann[6,7], Melanie Meyer[6] , Anaïs Carpentier[6] ,
Marc O Warmoes[1], Christian Jäger[1], Michel Mittelbronn[1,6,7,8,9,10], Jens C Schwamborn[1] ,
Maria Lorena Cordero-Maldonado[1], Alexander D Crawford[1,11,12], Emma L Schymanski[1] , Carole L Linster[1]

Juvenile neuronal ceroid lipofuscinosis (or Batten disease) is an autosomal recessive, rare neurodegenerative disorder that affects mainly children above the age of 5 yr and is most commonly caused by mutations in the highly conserved *CLN3* gene. Here, we generated *cln3* morphants and stable mutant lines in zebrafish. Although neither morphant nor mutant *cln3* larvae showed any obvious developmental or morphological defects, behavioral phenotyping of the mutant larvae revealed hyposensitivity to abrupt light changes and hypersensitivity to pro-convulsive drugs. Importantly, in-depth metabolomics and lipidomics analyses revealed significant accumulation of several glycerophosphodiesters (GPDs) and cholesteryl esters, and a global decrease in bis(monoacylglycero)phosphate species, two of which (GPDs and bis(monoacylglycero)phosphates) were previously proposed as potential biomarkers for *CLN3* disease based on independent studies in other organisms. We could also demonstrate GPD accumulation in human-induced pluripotent stem cell–derived cerebral organoids carrying a pathogenic variant for *CLN3*. Our models revealed that GPDs accumulate at very early stages of life in the absence of functional CLN3 and highlight glycerophosphoinositol and BMP as promising biomarker candidates for pre-symptomatic *CLN3* disease.

## Introduction

Neuronal ceroid lipofuscinoses (NCLs) are a group of rare neurodegenerative lysosomal storage disorders mainly characterized by the accumulation of hydrophobic autofluorescent material in the lysosomes (1, 2, 3). To date, more than 10 genes (designated *CLN1-14*) have been linked to different forms of NCL. In particular, mutations in *CLN3* have been associated with a subclass of NCLs termed juvenile NCL (JNCL) (2, 4) because of its onset between 5 and 8 yr of life. JNCL is characterized by progressive vision loss, intellectual delay, behavioral changes, epileptic seizures in some cases, and premature death (5). The most common mutation in patients is a deletion of 966 bp, referred to as 1-kb deletion, which leads to a major transcript of 521 bp and a minor transcript of 408 bp (6). The first encodes a truncated protein of 153 aa plus 28 novel aa, and the second encodes a protein product missing aa 154–263 of CLN3 because of in-frame skipping of exons 7–9 (6). Given the high conservation at the protein sequence level, models have been developed in several organisms to study the molecular and physiological functions of CLN3 and the mechanism underlying the associated disease (7, 8, 9, 10).

The *CLN3* gene encodes a transmembrane protein of 438 aa localized in the endolysosomal compartment (11, 12) and, at the functional level, has been mainly associated with pH homeostasis, protein trafficking, calcium signaling, phospholipid distribution, and lysosomal amino acid transport (9, 10, 13, 14). Despite extensive research efforts, the actual function of CLN3 in all these processes remains enigmatic and there is no effective treatment for *CLN3* patients that can prevent or at least slow down the disease progression. The main challenges for the development of new therapies are the low number of patients available for enrolment in controlled clinical trials and the scarcity of patient-derived samples. To progress in disease understanding and therapy development despite these challenges, four different *Cln3* mutant mouse strains have been developed so far. They recapitulate the classical cellular hallmarks of JNCL, including intracellular accumulation of autofluorescent material and lysosomal dysfunction (7, 8, 15, 16). All

[1]Luxembourg Centre for Systems Biomedicine, University of Luxembourg, Belvaux, Luxembourg    [2]Department of Environmental Health Sciences, Mailman School of Public Health, Columbia University, New York, NY, USA    [3]Max Planck Institute for Heart and Lung Research, Bad Nauheim, Germany    [4]Institut d'Investigació Sanitària Pere Virgili, Tarragona, Spain    [5]Biosfer Teslab SL, Reus, Spain    [6]National Center of Pathology (NCP), Laboratoire national de santé (LNS), Dudelange, Luxembourg    [7]Department of Oncology (DONC), Luxembourg Institute of Health (LIH), Strassen, Luxembourg    [8]Luxembourg Center of Neuropathology (LCNP), Dudelange, Luxembourg    [9]Faculty of Science, Technology and Medicine (FSTM), University of Luxembourg, Esch-sur-Alzette, Luxembourg    [10]Department of Life Science and Medicine (DLSM), University of Luxembourg, Esch-sur-Alzette, Luxembourg    [11]Department of Preclinical Sciences and Pathology, Norwegian University of Life Sciences (NMBU), Ås, Norway    [12]Institute for Orphan Drug Discovery, Bremerhaven, Germany

Correspondence: ursula.heins-marroquin@uni.lu; carole.linster@uni.lu

mouse models exhibit late onset of disease (6–12 mo of age) characterized by slow progressive neurodegeneration, motor dysfunction, loss of specific neuronal populations, microglial activation, and astrocytosis. However, there are notable differences between the strains in terms of the onset of phenotypic abnormalities and the affected neuronal populations, indicating that genetic background and the genetic manipulation strategy employed can modulate the phenotypic outcomes in different CLN3-deficient models (7, 15, 16, 17).

Because of the high cost and relatively late disease onset of rodent models, it is highly relevant to develop other models to speed up *CLN3*-related research and address the urgent needs of the patients. Human iPSC-derived cerebral organoids homozygous for the CLN3$^{Q352X}$ mutation were recently shown to recapitulate some of the NCL phenotypes (i.e., fingerprint-like intracellular inclusions, accumulation of subunit c of the mitochondrial ATP synthase [SCMAS], and lysosomal dysfunction) and represent thus a promising model to investigate the impact of CLN3 deficiency on brain development and neuronal function (18). In the last years, zebrafish emerged as a highly relevant and convenient whole-organism model to study developmental and mechanistic aspects of various human diseases (19). Its external development and optical clarity allow for monitoring the early development of organs such as the eye and the brain, which are among the most heavily affected organs in JNCL. A previous study using the morpholino technology to knock down *cln3* in zebrafish resulted in morphant larvae showing embryonic dysmorphology, lysosomal dysfunction, epileptic seizures, and premature death (20). In the present study, we tested different non-overlapping morpholinos against the *cln3* gene and we could not detect any obvious phenotypic changes. Moreover, we created two stable *cln3* mutant zebrafish lines using the CRISPR/Cas9 technology, and in agreement with *CLN3* knockout models in other species, we confirmed that this gene is not essential for early organismal development. In fact, homozygous mutants reached adulthood without showing any gross phenotype and were indistinguishable in size and overall morphology from wild-type siblings. However, a closer analysis of the locomotor behavior revealed that *cln3* mutant larvae show higher basal activity and higher sensitivity to the effects of pro-convulsive drugs. Contrary to the relatively subtle behavioral changes, LC-MS–based metabolomics and lipidomics analyses revealed very interesting and striking differences between wild-type and *cln3* mutant larvae even at the very early stage of 5 days post-fertilization (dpf). Glycerophosphoinositol (GPI) emerged as the most significantly elevated metabolite in the *cln3* mutant zebrafish larvae and bis(monoacylglycero)phosphate (BMP) 22:6 as one of the most significantly decreased lipid species. Importantly, we could demonstrate accumulation of certain glycerophosphodiesters (GPDs), including GPI, also in CLN3-deficient cerebral organoids derived from human-induced pluripotent stem cells, further supporting that GPD accumulation might start very early during brain development in the absence of functional CLN3. Our findings on increased GPD levels in CLN3-deficient models are in perfect agreement with recently published independent studies demonstrating elevations in these compounds also in CLN3-deficient yeast cells, mouse tissues, mouse and pig serum, and cultured human cells, human CSF, and human plasma (21, 22). All this makes GPI an attractive target candidate to be developed as a diagnostic biomarker for *CLN3* disease that so far does not exist. Moreover, GPDs can be used now as a reliable readout for CLN3 functionality and potentially for drug screenings.

In summary, Cln3-deficient zebrafish do not develop detrimental neurological defects but display behavioral phenotypes and robust and specific metabolic perturbations, also observed in mammalian CLN3-deficient systems, at very early developmental stages. Our stable *cln3* mutant zebrafish lines represent therefore a novel tool to make rapid progress in elucidating the physiological function of CLN3 and potentially screen for molecules that can rescue CLN3 function and benefit Batten disease patients.

# Results

## CLN3 conservation and expression in zebrafish

We found that the zebrafish genome encodes one *CLN3* ortholog candidate by blasting the human CLN3 protein (NP_001035897.1) against the zebrafish (taxid:7955) proteome using the BLASTP 2.8.0 tool from NCBI. Only Battenin or Cln3 (NP_001007307.1), a protein of 446 aa, showed a significant alignment to human CLN3, with a sequence similarity and identity of 65% and 51%, respectively, as shown previously (20). We extended the sequence analysis by including additional species and highlighting the residues associated with Batten disease in human CLN3 that are conserved in the zebrafish sequence (Fig S1A), providing a first argument in favor of zebrafish being a relevant organism for modeling this disease.

As there are no reliable antibodies for the detection of endogenous Cln3 protein in zebrafish, we analyzed *cln3* gene expression during embryonic development (6 hours post-fertilization to 7 dpf) using qPCR. We could detect *cln3* expression from the earliest time of measurement and found that transcript levels then progressively increased to finally stabilize after 4 dpf (Fig S1B), in good agreement with previously published in situ hybridization results (20). In addition, *cln3* transcript levels measured in various organs (brain, eye, heart, fin, skin, intestine, and gonads) of adult zebrafish females and males indicate ubiquitous expression with highest levels in the brain, eyes, and gonads (Fig S1C and D). These observations only partially overlap with human and mouse expression data, in which *CLN3* is expressed at similar levels in all the analyzed organs (23, 24). Although preliminary and pending confirmation at the protein level, these results suggest that, in zebrafish, *cln3* may play an important role more specifically in the brain, eyes, and gonads during adulthood.

## Transient knockdown of *cln3* in zebrafish

Because *cln3* is expressed at very early stages of zebrafish development, we first attempted to knock down gene expression using morpholino oligonucleotides (MOs) as a rapid approach to analyze the impact of Cln3 deficiency as compared to generating stable knockout lines. We injected 8 ng of a translation-blocking MO (TB-MO) or splice-blocking MO (SB-MO) and monitored the overall embryonic morphology up to 7 dpf (Fig 1A–C). We used two different

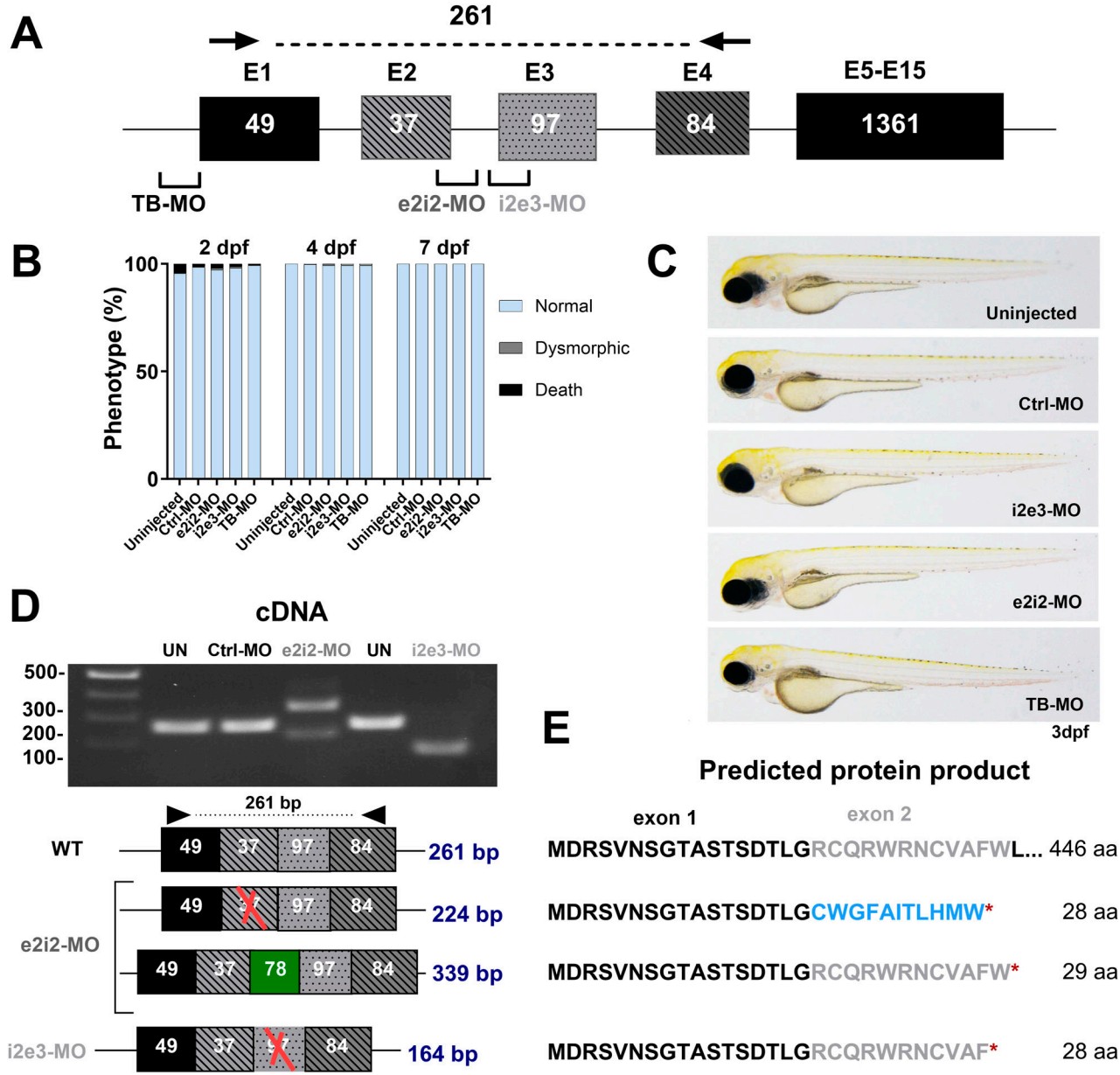

**Figure 1. Transient knockdown of *cln3* in zebrafish larvae.**
**(A)** Schematic illustration of the *cln3* gene showing the target sites for the translation-blocking (TB) and the two splice-blocking (SB; i2e3-MO and e2i2-MO) morpholinos (MOs) used in this study. Black arrows show the binding sites for the PCR primers used for SB-MO validation. **(B, C)** Microinjection of 8 ng SB- or TB-blocking MOs did not lead to any obvious morphological phenotype, and larvae displayed a normal development. The number of larvae analyzed at the indicated developmental time points: uninjected, 177; Ctrl-MO, 317; e2i2-MO, 250; i2e3-MO, 147; and TB-MO, 130. **(D)** Validation of MO efficiency by PCR amplification of target regions in cDNA of the *cln3* morphants in comparison with uninjected (UN) and control morpholino–injected larvae. **(E)** Schematic representation of transcribed sequences and resulting translation products after treatment with the SB-blocking morpholinos. Microinjection of i2e3-MO resulted in exon 3 skipping, leading to an early stop codon. In contrast, e2i2-MO treatment led to two different splicing events, both also resulting in a premature stop codon. Excluded exons are marked with a red cross, included intron in green, and new amino acid sequence in light blue.

SB-MOs that were designed to target the boundaries between exon 2 and intron 2 (e2i2), and intron 2 and exon 3 (i2e3), respectively (Fig 1A). Based on PCR amplification of the targeted exon–intron boundaries and subsequent analysis by agarose gel electrophoresis, we could confirm that for both SB-MOs, the wild-type band (261 bp) disappeared to be replaced by novel bands whose lengths were consistent with in silico predictions based on exon skipping or intron inclusion (Fig 1D). Transcript sequence alterations were confirmed by sequencing to result in premature stop codons encoding early truncated proteins of 28–29 aa, assuming the transcripts are stable enough to be translated (Fig 1E, Supplemental Data 1). Because of the absence of reliable Cln3 antibodies, the efficiency of the MOs could not be confirmed at the protein level. All three *cln3* morphants were morphologically indistinguishable from

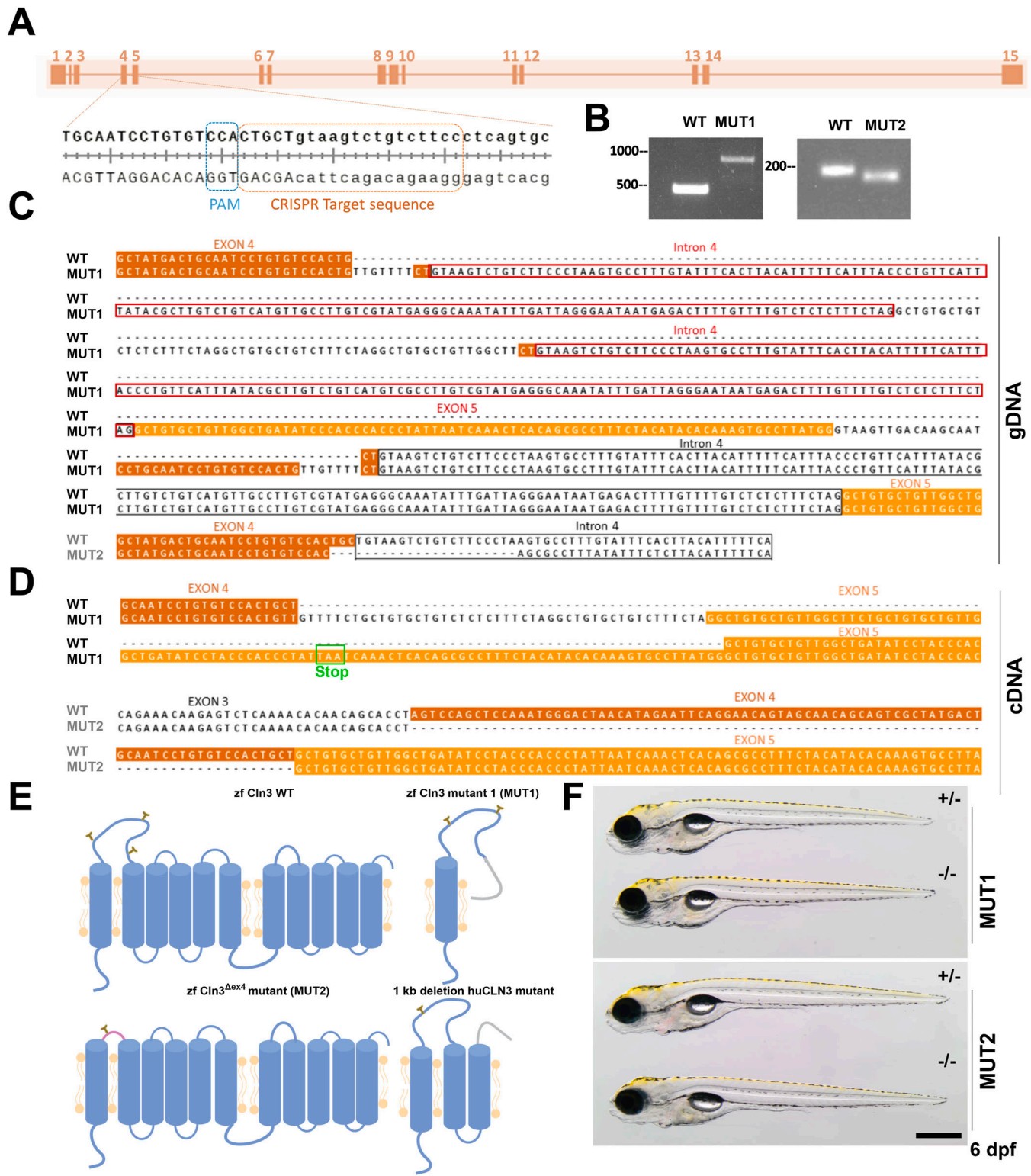

**Figure 2. Generation of two stable *cln3* mutant lines in zebrafish using CRISPR/Cas9.**
**(A)** Schematic map of the *cln3* gene with zoom on the gRNA target site. Uppercase letters represent the end of exon 4, and lowercase letters, the beginning of intron 4.
**(B)** Gel electrophoresis of PCR amplicons from the mutated region in gDNA of F2 founders. **(C)** gDNA sequences of the two stable *cln3* mutant lines generated by the CRISPR/Cas9 technology. **(D)** Based on cDNA sequence analysis from the pGEMT cloning, MUT1 carries an indel mutation resulting in a truncated translation product of 122 amino acids and MUT2 displays an in-frame deletion of exon 4. WT exon 4 sequence is highlighted in brown, exon 5 in orange, and intron 4 with a black box. Additional copies of intron 4 are highlighted with red boxes and the premature stop codon with a green box. **(E)** Schematic representation, based on the in silico prediction of transmembrane helices and N-glycosylation sites, of the expected protein products encoded by the WT, MUT1, and MUT2 alleles, as well as the main transcript of the most common human

their age-matched controls, as shown by representative individuals from the different groups at 3 dpf (Fig 1C). At higher morpholino concentrations, we observed toxicity in all embryos including those injected with the control morpholino. Altogether, the results strongly suggest that our knockdown approach is efficient and the absence of any obvious phenotype in the morphants indicates that *cln3* might not play an essential role during the early development in zebrafish.

### Generation of stable *cln3* knockout lines using CRISPR/Cas9

Mouse models for Cln3 deficiency showed a relatively late onset of symptoms (6–12 mo), including several hallmarks of Batten disease such as neurodegeneration and lipofuscin accumulation (6, 7, 16). Furthermore, in humans the disease only appears in childhood, suggesting that *CLN3* function may not be critical for morphogenesis or early development but rather may be involved in homeostasis, particularly in the brain, the organ most affected in the disease. Hence, we suspected that the potential phenotypes in zebrafish might also appear later in juvenile or adult individuals. Because morpholinos are typically efficient only until 5 dpf, studies in juvenile stages are not possible using morpholino-mediated approaches. Consequently, we generated two stable homozygous mutant zebrafish lines using the CRISPR/Cas9 technology by targeting the exon 4–intron 4 boundary of the *cln3* gene (Figs 2A and S2A–C). The first line (institutional code *cln3^{lux1}*, hereafter referred to as MUT1) carries a long indel mutation that inserts 496 extra bp in the genomic DNA (Fig 2B). The insertion contains two copies of intron 4 and one copy of exon 5 (Fig 2C). Analysis of the cDNA (based on the pGEMT cloning) revealed a main transcript encoding a truncated protein of 122 aa that contains 34 novel amino acids at the C-terminus (Fig 2D) and only includes the first transmembrane domain (Fig 2E). This transcript is even shorter than the main transcript detected in CLN3 patients with the most common 1-kb deletion mutation (Figs 2E and S3, and Supplemental Data 2) (6). Therefore, we presume that the resulting protein might have a complete loss of function and we considered the *cln3^{lux1}* mutant as a putative knockout line.

The second mutant line (institutional code *cln3^{lux2}*, hereafter referred to as MUT2) has a deletion of 22 bp at the genomic DNA level (Fig 2B and C) that disrupts the splicing process, resulting in an in-frame skipping of exon 4 at the cDNA level (first luminal loop of the transmembrane protein) and a predicted expression product that may retain residual function (Fig 2C–E, Supplemental Data 2). However, in both the human and zebrafish genes, exon 4 encodes two putative asparaginyl glycosylation sites (Fig S3) and contains a conserved serine residue mutated in some patients affected by Batten disease (25). Moreover, in a recent publication, it was demonstrated that CLN3 is mainly glycosylated in the lysosomes of ARPE19 cells (26), suggesting that disruption of glycosylation sites might affect the correct trafficking of CLN3 to its final destination. Using qPCR, we found that *cln3* mRNA levels were only slightly changed in MUT1 and MUT2 larvae compared with the respective

wild-type control larvae (Fig S2D), showing that the mutant transcripts do not undergo substantial nonsense-mediated mRNA decay. Finally, we confirmed by PCR and full-length cDNA sequencing that only one *cln3* transcript is present in WT, MUT1, and MUT2 larvae, indicating that no other splice variants are formed in our CRISPR mutant lines (Supplemental Data 2), although additional analyses such as long-read RNA sequencing would be needed to fully exclude this possibility. We decided to proceed with both mutant lines to study phenotypic impacts that may be relevant for human *CLN3* disease.

To determine whether homozygous *cln3* mutants are viable, we raised the progeny from F1 incrosses to adulthood and genotyped the fish after 2 mo. For both *cln3* mutant lines, 30–46% of the F2 progeny were homozygous mutants (Fig S2E), suggesting that Cln3 deficiency does not induce lethality at embryonic or juvenile developmental stages in zebrafish. Furthermore, homozygous mutants did not exhibit obvious differences in size, gross morphology, or fertility in comparison with their wild-type and heterozygous siblings (Figs 2F and S2F and G). Finally, survival was not affected in homozygous MUT1 fish compared with their heterozygous siblings (Fig S2H) and both lines were maintained as homozygous lines. The absence of mutations in the *slc45a2* gene, around the site targeted by the sgRNA co-injected with the *cln3* sgRNAs (to facilitate selection of successfully injected larvae in the P0 generation), was confirmed by genomic DNA sequencing in F2 and F3 generation animals (Supplemental Data 3).

### Behavioral assays in *cln3* mutant zebrafish larvae indicate early neurological defects

In zebrafish, the innate behavioral response depends on the correct development and functioning of the nervous system (27). To uncover potential neurological impairments in the *cln3* mutant larvae, we therefore performed three behavioral assays: global activity, dark–light response, and pro-convulsive drug response (Fig 3A). In the first approach, we studied the global activity of larvae in the dark or light condition, after a 15-min pre-adaptation period in the corresponding condition. In both conditions, MUT1 larvae showed a higher locomotor activity in comparison with heterozygous control larvae; for MUT2 larvae, a (non-statistically significant) higher basal activity was only observed in the light (Fig 3B and C). In the second approach, the lighting conditions were alternated in 15-min intervals (dark–light–dark), after a 15-min pre-adaptation period in the dark. In the first dark period, control larvae showed a constant velocity. During the light period, activity gradually increased, and in the subsequent dark period, larvae remained transiently hyperactive before recovering to a more passive swimming mode (Fig 3D). Mutant larvae displayed a similar dark–light behavioral response, except during the dark-to-light and light-to-dark switches (~min 15–20 and 30–32). During these transition periods, the control larvae showed abrupt changes in activity, with notably a rather dramatic freeze startle response upon turning on the lights, which was virtually absent in both the MUT1 and MUT2 larvae. The lack of this

Batten disease *CLN3* allele (1-kb deletion; *huCLN3*). **(F)** MUT1 and MUT2 mutants and heterozygous controls at 5 dpf. The scale bar represents 500 μm. Homozygous mutants are morphologically indistinguishable from their heterozygous counterparts. WT, wild type; zf, zebrafish; hu, human.

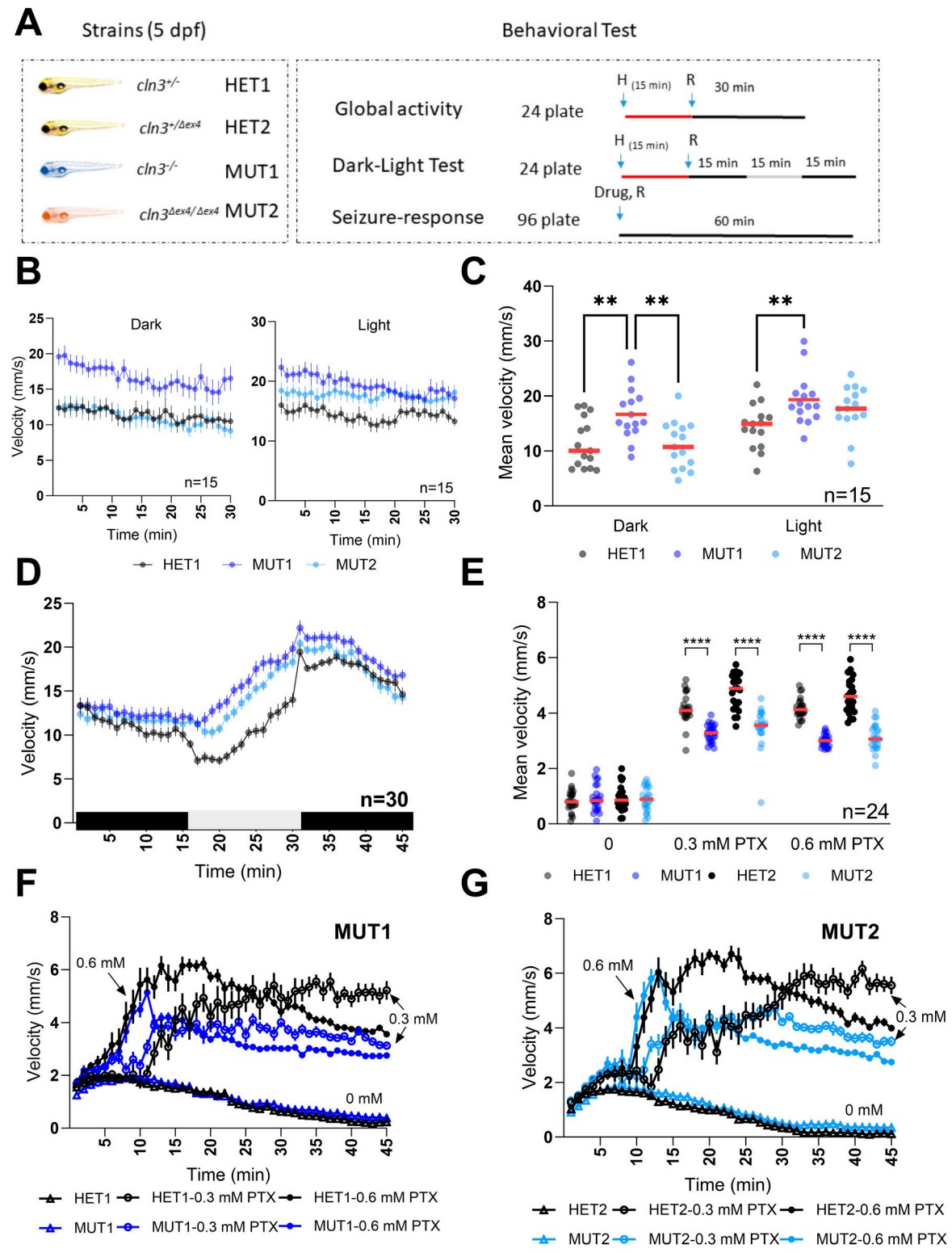

**Figure 3. Locomotor behavior of *cln3* mutant larvae under different light conditions.**
**(A)** Experimental design of behavioral response studies for 5 dpf larvae. For the global activity and dark–light response assays, 8 larvae/line were placed in a 24-well plate (1 larva/well) and pre-adapted for 15 min to the indicated condition (light/dark) before starting the recording. The movement was tracked for 45 min in 15-min light/dark intervals or for 30 min under one continuous lightning condition. For the seizure response assay, 12 larvae/line were placed in a 96-well plate and recording was started shortly after addition of PTX for 1 h. Vehicle control was 0.6% DMSO. **(B)** Mean velocity (1-min intervals) of the larvae recorded under constant lighting conditions (n = 15 for each line). **(B, C)** Total mean velocity recorded in panel (B). **(D)** Mean velocity (1-min intervals) of the larvae in alternating lighting conditions (n = 30 for each line). **(E)** Total mean velocity in 1-h exposure to 0, 0.3, and 0.6 mM PTX. **(E, F, G)** Behavioral profiles (1-min intervals) of the larvae recorded in panel (E) (n = 24 for each line). All

startle response upon the dark-to-light switch may result from early neurological impairment or decreased visual acuity in the Cln3-deficient larvae.

For the drug response assay, we exposed *cln3* mutant larvae to pentylenetetrazole and picrotoxin (PTZ and PTX, respectively). Both compounds are chemoconvulsant drugs that induce epileptic seizures in murine and zebrafish models (28, 29). First, MUT1 and heterozygous control larvae at 5 dpf (n = 12) were incubated with different concentrations of PTZ (5–20 mM) under dark conditions. After 10-min habituation in the dark, the locomotor activity was recorded for 1 h in that same condition. All tested PTZ concentrations induced an increase in activity in both the heterozygous and homozygous mutant *cln3* lines (Fig S4A and B). At subacute PTZ concentrations (7.5 and 10 mM), MUT1 larvae showed a significantly higher activity (2.01 ± 0.54 and 2.51 ± 0.39 mm/s) compared with the heterozygous control larvae (1.48 ± 0.34 and 1.94 ± 0.37 mm/s; values represent mean velocities moved during the 60 min). However, at acute PTZ concentrations (15 and 20 mM), the total mean velocities between control and MUT1 larvae were not significantly different (Fig S4A).

PTX was tested at 0.01, 0.03, 0.1, 0.3, and 1 mM using the same protocol as applied for the PTZ treatment (Fig S4C). A significant increase in locomotor activity was detected starting only at 0.3 mM PTX in the control heterozygous larvae, whereas 0.1 mM PTX already induced a significantly higher activity in MUT1 larvae. In contrast to the results obtained with PTZ and unexpectedly, at 0.3 and 1 mM PTX, MUT1 larvae showed a remarkably lower movement response (2.82 ± 0.53 and 2.13 ± 0.23 mm/s) compared with the heterozygous control larvae (4.32 ± 0.52 and 3.18 ± 0.62 mm/s) (Fig S4C and D).

The PTX phenotype was confirmed in further experiments that included also the MUT2 zebrafish line (Fig 3E–G). All larvae showed a similar behavioral profile in the vehicle condition (DMSO) (Fig 3F and G). In the presence of PTX, control and mutant larvae displayed a sharp increase in activity within 5–10 min. However, the activity of the *cln3* mutants plateaued at a significantly lower level after 10–20 min, compared with the heterozygous control larvae. In conclusion, these results showed that Cln3 deficiency does not lead to spontaneous epileptic seizures, at least in these early stages of zebrafish larval development. However, the behavioral phenotypes observed in the presence of different concentrations of proconvulsant drugs clearly indicated that Cln3 is important for healthy neuronal function and that some mild neurological impairments can already be observed in young (5 dpf) larvae.

### Metabolic perturbations in *cln3* mutant larvae detected by untargeted metabolomics

Because we observed an increased locomotor activity in the MUT1 line under normal conditions, we performed HILIC-HRMS–based untargeted metabolomics analysis in polar extracts of six pools of 40 MUT1 and 40 WT larvae at 5 dpf, obtained from different parental batches (Fig 4A). In this first experiment (hereafter referred to as

experiment 1), we detected 3,553 features (i.e., peaks) in a positive mode and 2,775 features in a negative mode (6,328 in total) within the range of 60–900 *m/z* (Table S1). Using a *t* test, we found that 29% of the features (i.e., 1,857) had differential abundance values (*P* < 0.05) between both lines. A total of 798 features (603 annotated) were lower in the mutant compared with WT larvae, whereas 1,059 features (766 annotated) were higher (Table S1 and Fig 4B). Principal component analysis using all measured features revealed a clear and consistent separation between WT and MUT1 samples (Fig S5A) suggesting profound differences at the metabolomics level. Lower clustering in this analysis for the MUT1 samples as compared to the WT samples suggested a certain degree of metabolic divergence in the mutant genotype.

Based on the metabolite enrichment analysis performed using MetaboAnalystR, carnitine synthesis, urea cycle, and several amino acid pathways (*e.g.*, "arginine and proline metabolism," "glycine and serine metabolism," and "aspartate metabolism") were ranked among the top perturbed metabolic pathways (Fig 4C). In addition, phosphatidylcholine biosynthesis, mitochondrial *β*-oxidation, and TCA cycle were identified as being significantly altered in MUT1 larvae. All these pathways have been previously highlighted in studies using different NCL models and patient-derived cells, supporting the disease relevance of our zebrafish model (13, 30, 18, 31, 32).

The most significantly changed metabolite in our comparative metabolomics analysis between WT and MUT1 larvae was glycerophosphoinositol (GPI), reaching ≥19-fold higher levels in the mutant extracts (Fig 5A and B). Adenosine diphosphate ribose was identified as the metabolite with the highest fold change decrease in MUT1, and glucosamine-6-phosphate was ranked as the (annotated) metabolite with the highest fold change increase in MUT1. Interestingly, we could also observe significant accumulation in other glycerophosphodiesters (GPDs) such as glycerophosphoglycerol (GPG; 1.4-fold) and glycerophosphocholine (GPC; threefold) (Fig 5B), whereas glycerophosphoethanolamine (GPE) was not significantly changed and glycerophosphoserine (GPS) was not detected in the extracts. Accumulation of GPDs has also been found in yeast and mammalian models of CLN3 deficiency (21, 22), suggesting that CLN3 plays a highly conserved role in maintaining normal GPD levels across species. Phospholipid precursors such as phosphocholine and phosphoethanolamine were also detected at significantly higher levels in the mutant compared with WT larvae (Fig 5B), reminiscent of a report of impaired phosphatidylethanolamine and phosphatidylcholine synthesis through the Kennedy pathway for *Saccharomyces cerevisiae* strains deficient in the CLN3 homologous protein Battenin or BTN1 (13). Consistent with the metabolite enrichment analysis, we observed a significant decrease in the levels of certain amino acids in the MUT1 larvae, including aspartic acid, glutamine, arginine, ornithine, and proline (Fig 5C). Furthermore, some amino acid–derived neurotransmitters appeared to be dysregulated in MUT1 mutants (Fig 5D). The levels of N-acetylaspartylglutamic acid (NAAG), the third most abundant neurotransmitter in the human brain

experiments shown are representative of at least three independent experiments performed on different days. In all the behavioral profiles shown, data points are means ± SEMs. In the scatter dot plots, each dot represents the total mean velocity of one larva and the red line represents the median. Statistically significant differences between lines were determined using the ordinary two-way ANOVA test followed by Sidak's multiple comparison test (**$P$ ≤ 0.0021 and ****$P$ ≤ 0.0001).

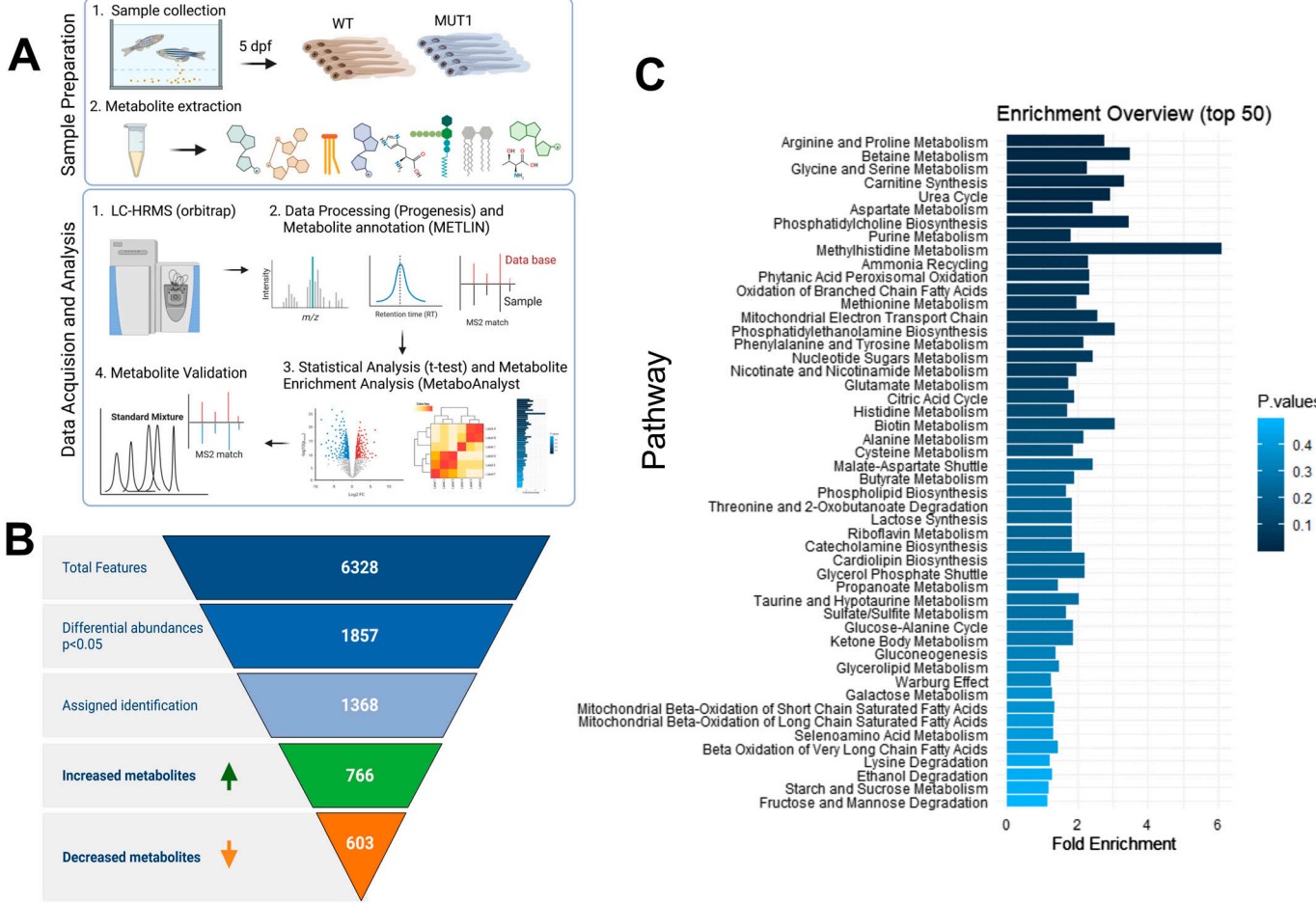

**Figure 4. Untargeted metabolomics differentiates between WT and MUT1 samples.**
**(A)** Experimental workflow of the untargeted metabolomics study from sample collection to data analysis. **(B)** MS feature filtering pipeline including statistical analyses and compound identification. Of 6,328 detected features detected in experiment 1, 1,857 differed significantly between the WT and MUT1 lines, and of those, 1,369 could be annotated. **(C)** Metabolite enrichment analysis of differential metabolites (*P* < 0.05) highlights the major dysregulated pathways between WT and MUT1 samples of experiment 1.

and previously proposed as a biomarker of severity in *CLN2* disease (33), were decreased about twofold in MUT1 larvae compared with WT (*P* < 0.0001). The inhibitory neurotransmitter γ-aminobutyric acid (GABA) unexpectedly showed slightly (10%) but significantly (*P* < 0.01) higher levels in MUT1 larvae. Interestingly, norepinephrine sulfate reached detectable levels in MUT1, but not in WT extracts. We also observed alterations in metabolic pathways that are not present in the Small Molecule Pathway Database used for our metabolite enrichment analysis. Notably, acylcarnitine (AC) synthesis appeared to be perturbed, with significantly decreased levels measured for most ACs detected in MUT1 larvae (Fig S5B), whereas the level of acetylcarnitine was slightly increased (Fig 5E).

The entire untargeted metabolomics analysis was repeated in a second independent experiment (experiment 2) using samples from different parental zebrafish pairs and yielded highly comparable results in terms of the total numbers of differential features and metabolite enrichment analysis (Fig S6A–C and Table S2), as well as most significantly changed single metabolites (Fig S7A and B to be compared with Fig 5). Finally, the identity of 16 statistically

significant differential metabolites (found in both independent experiments) was confirmed by direct comparison with the retention time and MS2 spectrum of reference standards (Table S3). We could not verify the identity of norepinephrine sulfate because of the lack of the corresponding reference standard.

Because we observed a reduction of proline levels in *cln3* mutants, we examined our differentially abundant metabolites for oligopeptides containing prolyl residues. We observed increased levels of these oligopeptide species in the MUT1 extracts in both of our data sets (Fig S8A), which, in combination with the decreased levels of free amino acids, may suggest a decreased lysosomal proteolytic activity in MUT1 larvae. One cellular phenotype observed in Batten disease is decreased cathepsin D (CtsD) activity (26), and CLN10 Batten disease is actually caused by mutations in CTsD itself (34). Moreover, it has recently been shown that cathepsin D (CtsD) maturation is impaired in CLN3 knockout ARPE19 cells (26). Although not statistically significant, we consistently measured lower CtsD activity in cln3 mutant whole-larva extracts, indicating lysosomal impairments that might be too mild to be detected at the peptidase

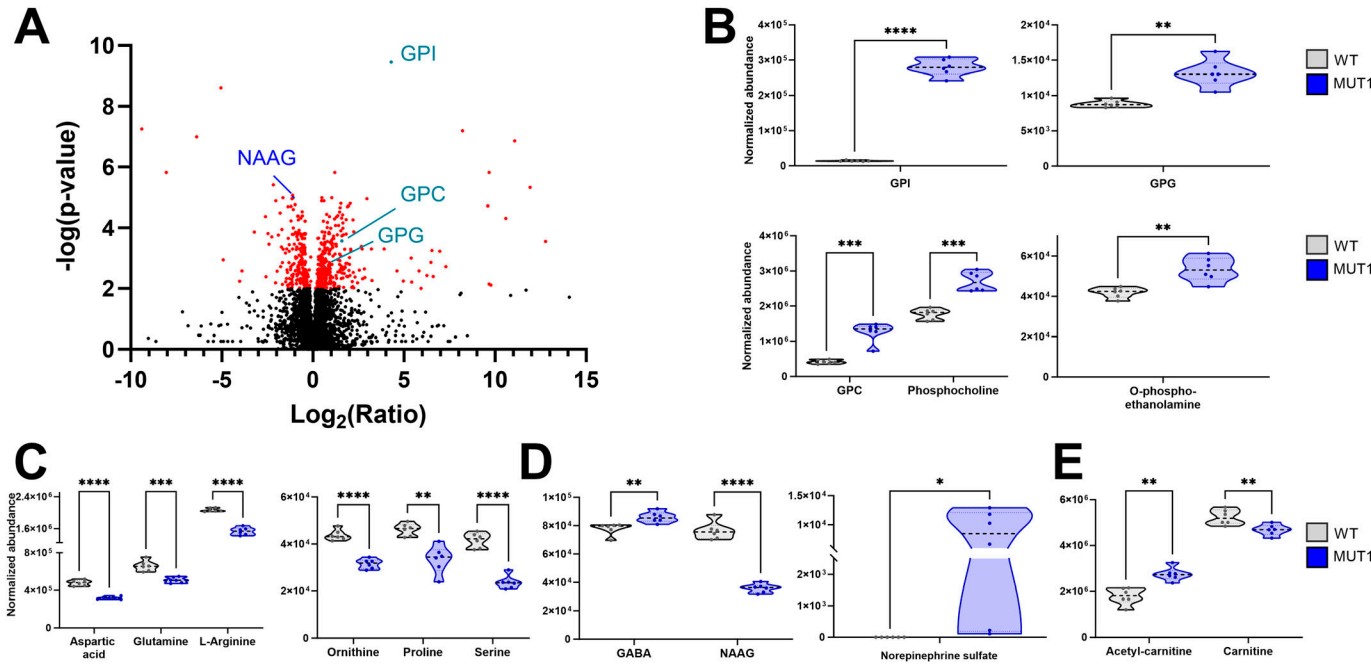

**Figure 5.  Most significantly altered metabolites between WT and MUT1 larvae based on untargeted metabolomics.**
**(A)** Volcano plot (corrected *P*-value versus fold change ratio) showing significantly (*P* < 0.001) altered metabolites in red for experiment 1. Each dot represents one metabolite. NAAG (previously reported as a potential NCL biomarker (33)) is highlighted in blue and GPD species in turquoise. **(B, C, D, E)** Violin plots showing levels of selected differential metabolites as abundance values, normalized as described in the Materials and Methods section. Statistically significant differences between the WT and MUT1 lines were determined using an unpaired multiple Welch's *t* test (**P* ≤ 0.05; ***P* ≤ 0.01; ****P* ≤ 0.001; and *****P* < 0.0001) (n = 240 for each line, extracted and analyzed in batches of 40).

level by our assay at this developmental stage (Fig S8B). CtsD protein levels were similar between the control and mutant extracts (Fig S8C). Finally, transmission electron microscopy in liver, intestine, and muscle tissues of 5 dpf MUT1 larvae did not reveal characteristic differences at the cellular level such as fingerprint-like inclusions, suggesting that this phenotype is not evident at early developmental stages, but should be further investigated in adult tissues (Fig S8D).

### Validation of metabolic disruptions using targeted analyses

As further validation of the untargeted metabolomics analysis, we performed targeted LC-MS–based measurements of GPDs, amino acids, and ACs. For GPDs that were not commercially available (GPI and GPE), the synthesis of standards was performed in-house as previously described by Laqtom et al. for GPE (21). For GPI synthesis, 25 mg of 1,2-diacyl-sn-glycero-3-phospho-(1-D-myo-inositol) was added to the reaction mixture instead of 1,2-dipalmitoyl-*sn*-glycero-3-phosphoethanolamine. New extracts were prepared from a new batch of larvae obtained from a different generation, allowing also to assess the conservation of the metabolic phenotypes across generations (Figs 6A and B and S9A and B). In full agreement with the untargeted results, we could confirm the accumulation of GPI, GPC, and GPG in MUT1 (Fig 6A and B). In addition, using the targeted method, we were able to detect GPS, which was also highly increased in the MUT1 larvae compared with WT1 larvae. The GPI and GPS peaks were not or barely detectable in WT samples

(Fig S9A). Further consolidating our results, we found that GPDs accumulate in a similar manner in MUT2 larvae compared with WT2 larvae, demonstrating that CLN3 function is also invalidated in this line. Interestingly, using our targeted LC-MS/MS method, we did not find accumulation of GPDs in a zebrafish model knocked out for ATP13A2/CLN12 (previously described in reference 35), suggesting that the abnormal GPD levels might be specific to CLN3 loss of function and not generalizable to all other NCL genes (Fig S9B). We also measured GPDs in human-induced pluripotent stem cell–derived cerebral organoids, a state-of-the-art human model for brain development previously described in references 18 and 36. Strikingly, GPI and GPG levels were threefold to 10-fold higher in organoids carrying the pathogenic CLN3$^{Q352X}$ variant compared with isogenic control organoids (Fig 6C). We also found an increase in GPS, but not in GPE and GPC levels in the mutant organoids. Taken together, our results strongly indicate that GPDs accumulate very early during (brain) development under CLN3 deficiency and that the role of CLN3 in maintaining normal GPD levels is conserved from fish to humans.

Finally, targeted amino acid analysis confirmed decreased levels of several amino acids in MUT1 and MUT2 (glycine, proline, isoleucine, leucine, threonine, and valine; Fig 6D), although only proline was consistently decreased in both the targeted and untargeted analyses. All other amino acids measured by our targeted method were either not significantly changed between wild-type and MUT samples or not consistent between both lines (Fig S9C). In contrast to the untargeted analysis, two independent

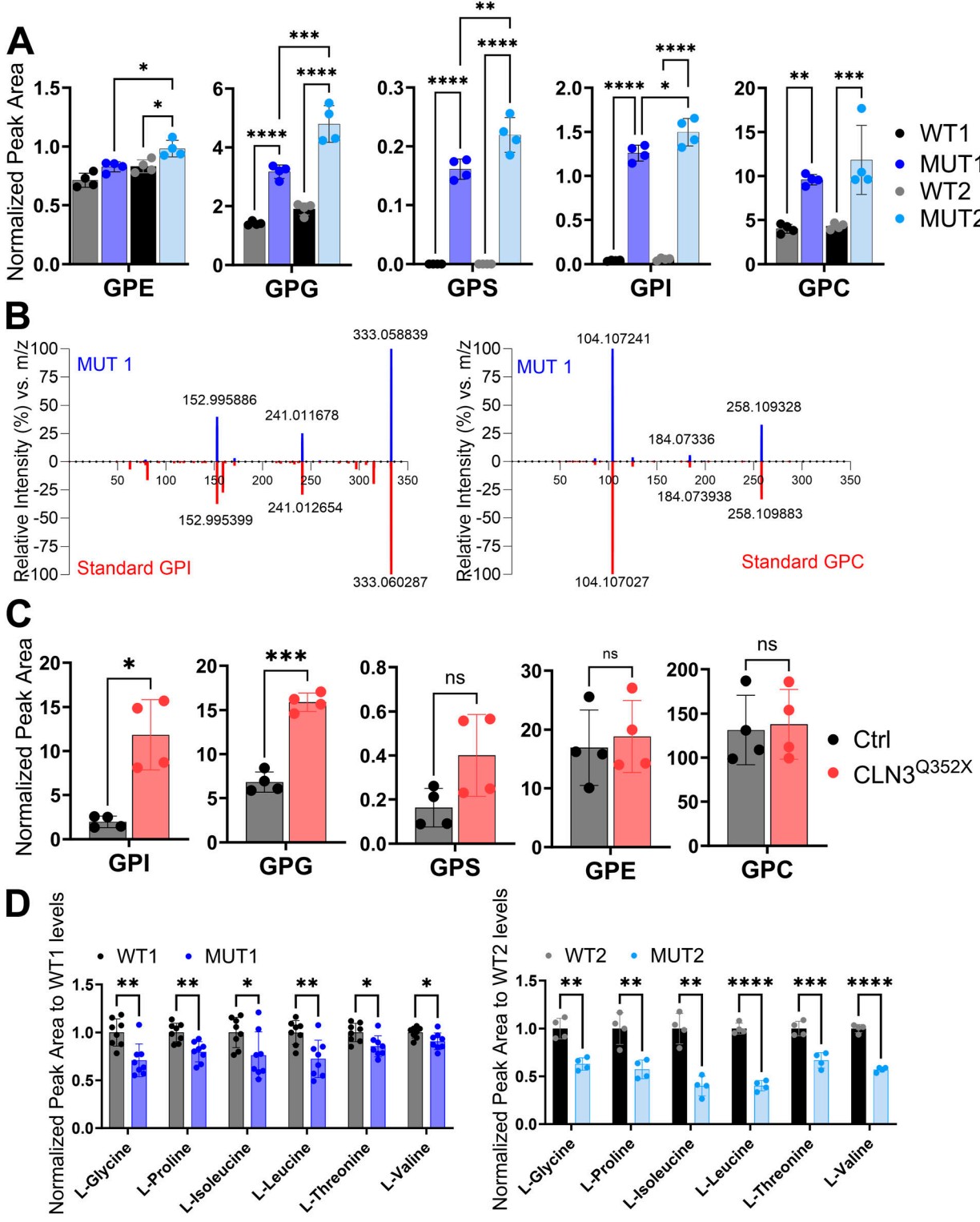

**Figure 6. Validation of glycerophosphodiester and amino acid changes by targeted LC-MS analyses.**
**(A)** Metabolite levels of glycerophosphodiester species (GPE, GPI, GPS, GPG, and GPC) extracted from batches of 10 whole zebrafish larvae at 5 dpf (n = 40 larvae for each genotype). **(B)** Representative MS/MS spectrum plots of GPI and GPC from MUT1 extracts (in blue) and metabolite standards (in red), further confirming compound identities (matching library score 83%, isotopic pattern score 100%). **(A, C)** Metabolite levels of the same glycerophosphodiester species as shown in panel (A), but extracted from human iPSC-derived cerebral organoids after 55 d of differentiation (n = 4). **(D)** Metabolite levels of amino acids differing significantly between WT and MUT1 zebrafish larvae at 5 dpf based on targeted LC-MS measurements. Data shown are means ± SDs of eight biological replicates for WT1/MUT1 (n = 200 larvae per genotype) and four biological replicates for WT2/MUT2 (n = 40 larvae per genotype). Statistically significant differences between WT and mutant zebrafish samples were

targeted methods showed slightly increased carnitine and acyl-carnitine levels in MUT1 versus wild-type samples (Fig S10). Given that the targeted and untargeted analyses were performed on samples derived from larvae belonging to different generations, it is difficult to conclude at this stage whether any discrepancies observed between the two approaches have a methodological or biological origin. Many metabolic changes were in agreement between the two methods, and those should be considered as the most robust Cln3 deficiency–induced changes that can be detected in zebrafish larvae, in a generation-independent manner. It remains to be established whether acylcarnitine levels are consistently affected by this genetic deficiency in zebrafish as well.

### Further evidence for perturbations in lipid metabolism in *cln3* mutant larvae by lipidomics

We performed HRMS-based untargeted lipidomics analyses on the non-polar extracts of both experimental sets and detected 11,447 MS features (4,927 in a negative mode and 6,520 in a positive mode), of which a total of 1,901 were annotated based on MS2 spectra (181 in a negative and 1,720 in a positive ionization mode) (Table S4). We identified 39 lipids with a fold change >2 and $P < 0.001$. Diverse lipids from different classes (Table S5), including glycerolipids, were altered in MUT1 larvae (Fig 7A and B). In particular, bis(monoacylglycero) phosphate (BMP 22:6_22:6) and acyl steryl glycoside (ASG 27:1;O;Hex; FA 14:1) displayed highly significant changes (>threefold). To confirm our results, we performed targeted lipidomics analysis of 20 different lipid classes in MUT1 and WT larvae using LC-MS in the multiple reaction monitoring mode. This targeted analysis confirmed a global decrease in BMP species at 5 dpf in MUT1 larvae, with the highly abundant BMP 22:6_22:6 showing the most important -fold change (Fig 7C). Interestingly, whereas individual cholesterol ester species (CE) changes between MUT1 and WT1 larvae were not significant, the summed level of cholesterol esters was significantly increased in MUT1 larvae (Fig 7D), being in perfect agreement with results obtained in isolated late endosome/lysosome compartments from the frozen human prefrontal cortex tissue (37). Furthermore, triacylglycerides, and hexosyl- and lactosylceramides also showed (non-significant) elevations in MUT1 larvae. In future work, it will be interesting to test whether these early trends for certain lipid classes become more pronounced at later developmental stages. We did not observe any significant changes for ceramides or dihydroceramides, indicating that only the glycosylceramides are affected, which is in line with the acyl steryl glycoside accumulation that was observed in the untargeted analysis. Steryl glycoside formation in animals has been reported to be catalyzed by glucocerebrosidases via transglycosidation of glycosylceramides (38, 39). Finally, no significant differences were observed for diacylglycerols (DG), plasmalogens (PL), sphingomyelins (SM), phosphatidylcholines (PC), phosphatidylethanolamines (PE), phosphatidylglycerols (PG), phosphatidylinositols (PI), phosphatidylserines (PS), lysophosphatidylcholines (LPC), and lysophosphatidylethanolamines through the targeted analyses (Fig S11).

## Discussion

In the present study, we aimed at investigating the function of *CLN3* using zebrafish as an in vivo model. In the first approach, we knocked down *cln3* using three different morpholino oligonucleotides (one translation-blocking MO and two splice-blocking MOs), but an obvious embryonic phenotype could not be observed in any of the generated morphants after injection of subtoxic MO doses. This contrasted with the severe phenotypes previously published in another study (20) in which similar MO target sites on the *cln3* gene were selected, although the exact MO sequences and the genetic background of the zebrafish strains differed. Although we used the AB strain for our studies, Wager and colleagues (20) used the TLF (Tüpfel long fin) strain. The latter is also a commonly used WT line that carries two known mutations, *leo*[t1] and *lof*[dt2], leading to changes in the pigmentation pattern and the development of long fins, respectively. Although studies in mouse models of *CLN3* disease have highlighted the relevance of the genetic background aspect on phenotype development (17), it seems unlikely that solely background effects could lead to such divergent results as observed here and previously. In the Wager study, a low dose of the TB-MO led to strong embryonic phenotypes, including abnormal brain and heart morphology, epileptiform activity, impaired survival, abnormal motor behavior, neuropathological features, and increased lysosomal storage. Although, as for the Wager study, the efficiency of our TB-MO could not be validated because of the absence of specific antibodies against zebrafish Cln3, we could confirm that both of our splice-blocking MOs very efficiently interfered with normal *cln3* transcript generation. The SB-MO used by Wager et al. (also validated for effective splicing interference via RT–PCR and sequencing) had to be injected at a more than sevenfold higher dose to cause the morphological abnormalities that they observed with the TB-MO, and the entire remaining study was therefore conducted solely with the (non-validated) TB-MO. Wager et al suggested that the lower efficiency of the SB-MO to induce a phenotype may be due to maternal *cln3* mRNA deposition (maternal mRNA is fully spliced and therefore not generally affected by SB-MOs) (40). In our study, we can exclude any kind of parental contribution in both mutant lines, as embryos were obtained by incrossing of *cln3* homozygous pairs, in which no fully functional *cln3* transcript should be present. Finally, although morpholinos are a great tool for functional genomic studies during zebrafish development, these types of experiments should be carefully controlled for off-target effects through rescue studies or MO injection into embryos homozygous for a null allele or an allele lacking the MO-binding site (41), especially when observing severe phenotypes.

As both our translation- and splice-blocking MO treatments remained phenotypically silent, to circumvent the ever-lingering doubts about off-target effects surrounding the studies that are exclusively MO-based, we generated stable *cln3* mutant lines using

---

determined using multiple comparison one-way ANOVA followed by Tukey's correction, and for iPSC samples, an unpaired *t* test was used (*$P \leq 0.05$; **$P \leq 0.01$; ***$P \leq 0.001$; and ns, not significant).

**Figure 7. Lipidomics analysis in WT and MUT1 larvae.**
**(A)** Volcano plot with annotated lipids found by untargeted lipidomics in non-polar extracts derived from the zebrafish larva samples generated in both experimental sets as described in the main text. Each dot represents a metabolite, and lipids with a significantly different abundance value in WT and MUT1 extracts are shown in red ($P < 0.001$). Gray dotted lines indicate the 1.5-fold change $\log_2$ ratio cutoff. BMP (44:12) and ASG (27:1; O;Hex;FA 14:1) are highlighted in dark and light blue, respectively. **(B)** Violin plots for the most significantly altered lipids based on untargeted lipidomics analysis. **(C)** Targeted BMP analysis in zebrafish extracts at 5 dpf. Data shown are means ± SDs for six biological replicates. Statistically significant differences between WT (gray) and MUT1 (blue) were determined using an unpaired multiple Welch's *t* test (*$P \le 0.05$; **$P \le 0.01$; ***$P \le 0.001$; and ****$P \le 0.0001$). **(D)** Stacked bar graphs for the indicated lipid classes measured by targeted lipidomics in 5 dpf zebrafish larvae. Each stacked bar is the sum of the average amounts measured for the indicated lipid species in six biological replicates of a pool of 40 larvae, normalized against its DNA

the CRISPR/Cas9 technology. In agreement with our morpholino results, both CRISPR homozygous *cln3* mutant lines generated here displayed normal development and reached adulthood without any obvious phenotype. Our observations are also consistent with results obtained in murine models, in which Cln3 deficiency was not lethal ([17]). We cannot exclude that the *cln3* transcript variants expressed in the SB-MO–treated morphants and in the two generated CRISPR lines may conserve some residual function, as it was reported for the *Cln3* murine models and the most common 1-kb deletion variant found in JNCL patients ([6], [42]). However, the truncated protein predicted to be expressed in the MUT1 line contains only the first transmembrane domain and is much shorter than the predicted main expression product of the 1-kb deletion variant. The predicted MUT2 expression product misses glycosyl-ation sites, which seem to be important for lysosomal trafficking ([25], [26]). Therefore, we believe that we have potentially generated loss-of-function mutations in both MUT lines. We could not detect a major decrease of *cln3* mRNA levels in either mutant, reducing the chance that transcriptional adaptation mechanisms (triggered by mutant mRNA decay) are compensating *cln3* loss-of-function phenotypes in our zebrafish model ([43]). Finally, the very distinct accumulation of GPDs and a decrease in BMP species in MUT1 and/or MUT2 larvae confirm that we have indeed successfully impaired Cln3 function, as the exact same metabolic changes were measured in samples derived from patients with *CLN3* disease ([21], [22], [44]). The association between BMPs and CLN3 has been known for many years, with metabolic labeling studies showing early on that CLN3 may be involved in BMP synthesis ([44]). In contrast, the accumulation of GPDs in CLN3-deficient mammalian cells and tissues and in extracellular fluids (CSF and plasma) of CLN3 patients was only shown very recently in two independent studies ([21], [22]). It is remarkable that we were able to detect GPD accumulation by untargeted metabolomics in an unbiased way and in extracts prepared from whole 5 dpf larvae. In tissue extracts of the Cln3 knockout mouse model, GPD accumulation became much more evident in samples enriched in lysosomes (lysoIP approach) and was shown for ~7-mo-old animals ([21]). In this latter study, evidence is also presented for cultured cells that CLN3 function is required for efflux of GPDs from the lysosomes to the cytosols, where they are normally further catabolized during phospholipid catabolism ([21]).

BMP is enriched in the inner leaflet of endolysosomal membranes, and its principal function remains unclear. Also, little is known about the pathways involved in BMP formation and degradation, although its synthesis is predicted to occur in the late endosomes/endolysosomes, as its concentration increases over the maturation of the latter ([45]). BMP is highly enriched in intra-luminal vesicles of multivesicular bodies, which are involved in cholesterol efflux and sphingolipid degradation ([45]). Accordingly, BMP seems to play a key role in lipid metabolism and membrane composition by facilitating the adhesion of soluble, positively charged hydrolases and activator proteins such as sphingolipid activator proteins SapsA-D and the GM2 activator protein to

intracellular membranes ([46]). It is therefore possible that the BMP depletion in CLN3-deficient cells also negatively affects the function of glucocerebrosidases, which may then possibly explain the increased levels of glycosylceramide and acyl steryl glycoside species that we observed in MUT1 larvae ([47]). Moreover, BMPs have a cone-like shape that affects membrane curvature, playing an important role in endosomal vesicularization and hence also in the maturation of late endosomes ([46]). In summary, CLN3 deficiency leads to profound changes in the lipidome, particularly in lipids associated with the endolysosomal compartment (BMP depletion and GPD accumulation), already during early development, which may compromise endolysosomal membrane integrity and transport mechanisms across these membranes, as well as cause further perturbations in endolysosomal metabolic activities, likely to lead progressively to dysfunctional lysosomes accumulating also other lipids (e.g., cholesterol esters, triglycerides, and glycosyl- and lactosylceramides) to abnormal levels ([Fig 8]).

Although the metabolic perturbations that we found in our *cln3* mutant zebrafish larvae are perfectly in line with findings in samples derived from patients with *CLN3* disease, the behavioral phenotypes observed in these larvae were surprisingly subtle when compared to the severe neurological symptoms that are, unfortunately, commonly observed in the patients. However, the changes that we could find were robust and, given that we observe them very early in development, may help shed light on critical neuro-developmental processes that are impaired in CLN3 patients, well before the symptoms manifest. Locomotor behavioral assays at 5 dpf showed that the MUT1 larvae have higher basal activity, in both light and dark conditions, compared with control larvae. In contrast, the MUT2 line showed only (non-significantly) increased activity in the light condition. In the dark–light response assay, both homozygous mutant lines showed a much less pronounced startle response upon the lightning change. Considering that visual impairment is the first obvious symptom to manifest in juvenile BD patients ([48]), it will be very important to clarify whether this phenotype has a retinal or neuronal origin.

Studies in *Cln3^{lacZ/lacZ}* mice revealed that they are more prone to develop seizures when treated with subacute doses of the chemo-convulsant PTZ ([8]). We tested PTZ in 5 dpf zebrafish larvae, and we observed a significant increase in locomotor activity in the MUT1 line compared with control larvae at subacute concentrations. Interestingly, an even more pronounced phenotype was observed upon exposure to PTX. Our results suggest that *cln3* mutants are more sensitive to PTX, because MUT1 larvae showed hyperactivity already at 0.1 mM, whereas in control larvae, the hyperactivity only occurred starting at 0.3 mM. Strikingly, at PTX concentrations higher than 0.3 mM, the mutant activity significantly dropped compared with the wild-type larvae. As described for PTZ ([49]), wild-type behavior under exposure to PTX can be divided into three stages. In the first minutes, the swimming activity increases (stage I, ≈ 0–10 min), followed by a rapid "whirlpool-like" circling swimming behavior (stage II, ≈ 10–26 min) and finally a progressive decrease in the activity (stage III, ≈ 26–60 min). The latter stage is characterized by clonus-like convulsions and loss of posture

concentration. Error bars represent SDs. Statistically significant differences between both zebrafish lines were determined using a two-way ANOVA test on the summed averages for each lipid class (**$P \leq 0.01$) (n = 240 larvae for each genotype).

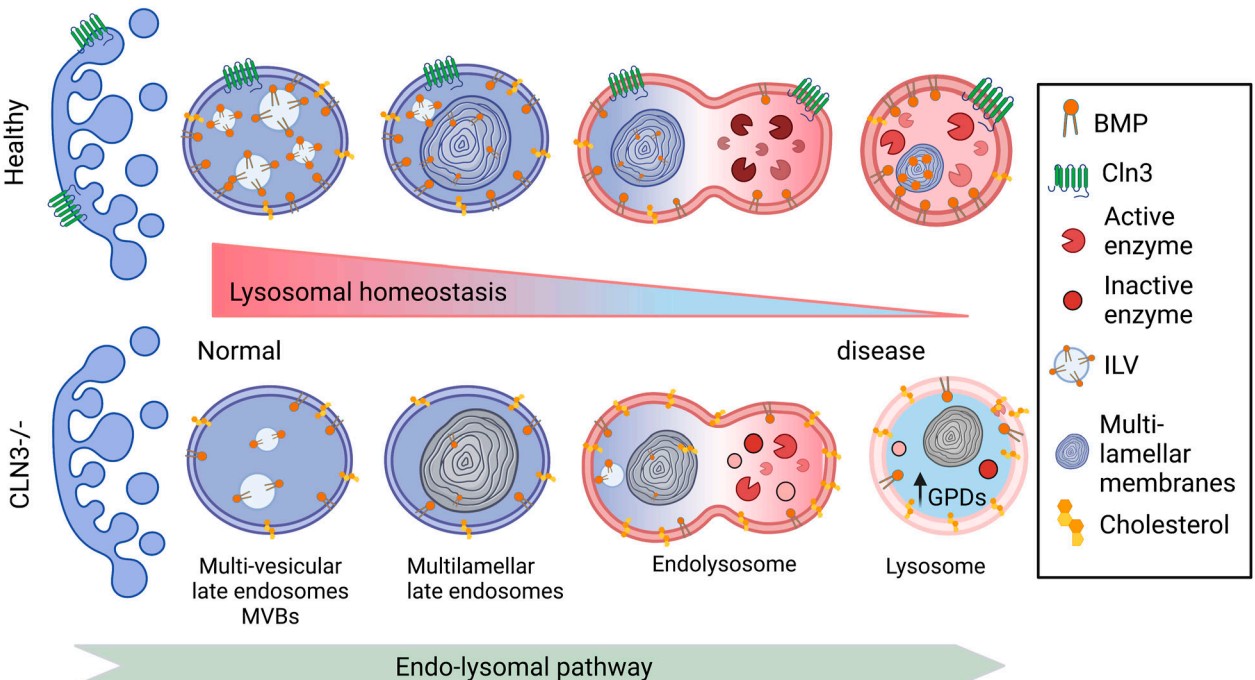

**Figure 8. Proposed changes induced by CLN3 deficiency along the endolysosomal pathway.**
In healthy cells, the number of intraluminal vesicles (ILVs) increases in the early endosomes to form multilamellar bodies in the late endosomes. The latter fuse with lysosomes forming a hybrid transient endolysosome to finally form secondary lysosomes. BMP is enriched in ILVs, and its levels increase during endosome maturation, contributing to ILV formation, ceramide degradation, and cholesterol efflux (45). In CLN3-deficient cells, BMP levels are decreased, potentially leading to reduced ILV formation, as well as impaired degradation and therefore accumulation of cholesterol, glycosylceramides, and glycerophosphodiesters, all contributing to lysosomal dysfunction.

that unfortunately cannot be accurately detected by the automated behavioral tracking system that we used. However, the decrease in the movement that we observed in our experiments certainly suggests that the larvae had entered stage III. Strikingly, both *cln3* mutants clearly showed stages I and III, but stage II seemed to be very short or completely absent. One interpretation is that the *cln3* mutant larvae skip most of the stage II reaction ("whirlpool-like" circling swimming) and directly enter stage III, in which the larvae undergo repeated episodes of convulsions (49). In this stage, the larvae display clonus-like seizures reflected by a rapid movement followed by postural loss where larvae turn upside down and remain immobile (49). The skipping of stage II would explain the decrease in the total movement that we observed with both *cln3* mutant lines.

No obvious abnormalities were observed in the MUT1 and MUT2 lines at the adult stage, which may be explained by the remarkable ability of zebrafish to regenerate neuronal cells (50, 51) and/or the induction of other compensatory mechanisms. In contrast to mammals, the zebrafish adult brain is enriched with a wide variety of neurogenic niches that constantly generate new neurons (52). Moreover, inflammation caused by brain injury or by induced neurodegeneration activates specific neurogenic programs that efficiently overcome the loss of neurons, allowing the maintenance of the tissue architecture in zebrafish brain (53). This regenerative potential of zebrafish may mask more pronounced loss-of-function phenotypes expected to develop with an increasing age in CLN3 deficiency.

Altogether, our MUT larvae recapitulate, within the first 5 d of life, metabolic abnormalities observed at a more advanced age in other

CLN3 models and patients, provide further support for perturbed lipid metabolism in the pathogenesis of *CLN3* disease, and highlight BMPs and GPDs as promising biomarker candidates for very early detection of this disease, maybe long before symptoms manifest. Our findings at larval stages open new avenues for investigation of the physiological function of CLN3 in an in vivo model that is more time- and cost-efficient than other whole-organism models. The observation that GPDs accumulate to high levels in CLN3-deficient zebrafish larvae and human cerebral organoids, that is, during early developmental stages, suggests that GPD metabolism is very tightly connected to CLN3 function. The absence of GPD accumulation in our zebrafish model for ATP13A2/CLN12 deficiency (35), also leading to neuronal lipofuscinosis (54, 55, 56), indicates that these compounds may be specific for certain types of NCLs, a hypothesis that will now need to be tested in different models and non-CLN3 Batten disease patients. The zebrafish model has the potential to fill the gaps between cell culture and mammalian models and accelerate the understanding of early metabolic events that lead to the development of a childhood disease that, as of today, is still invariably lethal.

## Materials and Methods

### Ethics statement

The Zebrafish Facility at the Luxembourg Centre for Systems Biomedicine is registered as an authorized breeder, supplier, and user

Developmental models of CLN3 deficiency    Heins-Marroquin et al.    https://doi.org/10.26508/lsa.202302057    vol 7 | no 3 | e202302057    **15 of 22**

of zebrafish with Grand-Ducal Decree of 20 January 2016 and 26 January 2023. All practices involving zebrafish were performed in accordance with European laws, guidelines, and policies for animal experimentation, housing, and care (European Directive 2010/63/EU on the protection of animals used for scientific purposes). Authorization number LUPA 2019/01 allowed the generation of the *cln3* mutant lines in this project, and authorization number LUPA 2017/04 allowed the performance of fin biopsies for genotyping purposes. Experiments using larvae older than 5 dpf were performed under Grand-Ducal Decree of 10 December 2012.

### Zebrafish lines and handling

Wild-type zebrafish (AB) and mutant lines (*nacre*, *cln3*[lux1], *cln3*[lux2], and *atp13a2*[sa18624]; all in the AB background) were kept in the Aquatic Facility of the Luxembourg Centre for Systems Biomedicine according to the standard protocols (57). Zebrafish embryos were obtained by natural spawning and reared in 0.3X Danieau's solution (17 mM NaCl, 2 mM KCl, 0.12 mM $MgSO_4$, 1.8 mM $Ca(NO_3)_2$, 1.5 mM Hepes, pH 7.5, and 1.2 $\mu$M methylene blue) in 10-h dark–14-h light conditions at 28°C. Adult fish were maintained in 3.5-liter tanks at a stock density of 4–6 fish/liter with the following parameters: water temperature, 28.5°C (±0.5); light:dark cycle, 14 h:10 h; pH 7.5; and conductivity, 700–900 µS/cm. Fish were fed three to five times a day, depending on age, with granular and live feed (*Artemia salina*). For live imaging, larvae were first anesthetized with buffered 0.008% tricaine methane sulfonate (MS222) and then embedded in 2.5% methylcellulose. Body length and eye area were analyzed using ImageJ (v1.53q). For locomotor behavioral studies, homozygous mutant embryos were obtained by incrossing MUT1 or MUT2 homozygous adult zebrafish and the heterozygous control larvae were generated by outcrossing homozygous MUT1 or MUT2 adults with the respective WT controls. For untargeted and targeted metabolomics analyses, MUT1 or MUT2 animals and the corresponding WT control lines were incrossed for generation of the respective homozygous larvae. An overview of the generations used in each experiment for the mutant lines is provided as a table in Supplemental Data 4.

### Morpholino antisense oligonucleotide microinjection

Translation-blocking (TB) and splice-blocking (SB) morpholinos (MOs) for *cln3* were purchased from Gene Tools with the following sequences (coding nucleotides are shown in uppercase letters): CLN3_SB-e2i2, 5′-gattgactcaacattaccAGAACGC-3′; CLN3_TB-ATG, 5′-TCGATCCATtgcgactttcacaggA-3′; and CLN3_SB-i2e3, 5′-CCCCAGCAAcctaaacagagataat-3′. A standard MO with the sequence 5′-CCTCTTACCTCAGTTACAATTTATA-3′ was used as a non-targeting MO control. All morpholinos had a 3′-lissamine or 3′-fluorescein modification. Stock solutions (2 mM) were prepared according to the specifications of the provider, and titrated working solutions were freshly prepared for each experiment. Manual microinjections of MOs were performed as previously described (58). Briefly, gene knockdown was achieved through microinjection of 1 nl of *cln3*-MO or control-MO into the yolk of wild-type embryos at one- to two-cell stage using an Eppendorf FemtoJet 4X microinjector. Injected embryos were incubated at 28°C, and

fluorescently sorted and evaluated daily until needed for the analytical experiments.

### CLN3 sequence alignment

Multiple sequence alignment was performed with MUSCLE software from EMBL-EBI using the following CLN3 protein sequences: *S. cerevisiae* (NP_012476.1), *Danio rerio* (NP_001007307), *Homo sapiens* (NP_001273038), and *Mus musculus* (NP_001139783). The Jalview software was used for editing the alignment for visualization purposes.

### Generation of CRISPR/Cas9 mutants

#### sgRNA template generation and transcription
The target sequences in the *cln3* and *slc45a2* genes were selected using the online tool CHOPCHOP (59). For generation of the sgRNA templates for *cln3* and *slc45a2*, 60-nt gene-specific oligos were designed. Both oligos contained a T7 (5′-TAATACGACTCACTATA-3′) promoter sequence, a 20-nt target site without the protospacer adjacent motif, and a 23-nt complementary region to the tracrRNA tail. The oligos were annealed to a generic oligonucleotide encoding the reverse complement of the tracrRNA tail. Subsequently, the ssDNA overhangs were filled in with T4 DNA polymerase (NEB) following the manufacturer's instructions. The resulting templates were purified using the Wizard SV Gel and PCR Clean-Up System kits (Promega) and transcribed using the MEGAscript T7 kit (Thermo Fisher Scientific) according to the manufacturer's instructions. The transcription reaction was performed for 4 h, and the preparation was then treated with 1 $\mu$l TURBO DNase for 15 min at room temperature. The sgRNAs were precipitated with ammonium acetate and resuspended in 50 $\mu$l water, and the concentration was determined by measuring $A_{260}$ (NanoDrop 2000; Thermo Fisher Scientific). All nucleotide and primer sequences used in this study are listed in Table S6.

#### sgRNA and Cas9 protein microinjection
200 ng of *cln3* sgRNA (0.5 $\mu$l), 200 ng of *slc45a* sgRNA (0.5 $\mu$l), 160 ng of Cas9 Nuclease NLS from *Saccharomyces pyogenes* (1 $\mu$l, NEB), and 0.2 $\mu$l phenol red were mixed in an Eppendorf tube (total volume 2.2 $\mu$l) and incubated for 5 min at room temperature. The mixture was placed on ice, and 1 nl was injected into one-cell embryos (58). *Slc45a2* sgRNA was used to track successful injections with albino clones, indicating that Cas9 had worked efficiently. Chimeras showing an albino phenotype were selected and raised in the LCSB Aquatic Facility as described above.

#### PCR analysis from whole larvae
Larval genomic DNA was prepared using the HotSHOT protocol (60). Briefly, single injected 1 dpf embryos were dissociated in PCR tubes containing 10 $\mu$l of 50 mM NaOH at 95°C for 20 min. After cooling down, 1 $\mu$l Tris–HCl, pH 7.5, was added and the preparation was mixed. 1 $\mu$l of this mixture was used as a genomic DNA template in PCRs using the primers listed in Table S6. Amplicons were analyzed on 2.5% agarose gels prepared with 1x Tris–acetate–EDTA buffer (Sigma-Aldrich).

### PCR analysis from adult fish by fin biopsy

Adult zebrafish were anesthetized in 0.016% MS222. A small portion of the caudal fin was cut, placed in an Eppendorf tube containing 250 $\mu$l lysis buffer (100 mM Tris–HCl, pH 8, 200 mM NaCl, 0.2% SDS, and 5 mM EDTA, pH 8) supplemented with 0.1 mg/ml proteinase K, and digested at 55°C for 2 h or overnight with vigorous shaking. Proteinase K was inactivated for 20 min at 90°C, and 400 $\mu$l of isopropanol was added to the cooled samples, followed by vortexing and centrifugation at 13,000$g$ and 4°C for 30 min. The pellet was washed with 100 $\mu$l of 70% ethanol and then dried at 55°C for 10 min. The washed pellet was resuspended in 50 $\mu$l water and finally incubated for 10 min at 55°C to dissolve the DNA. The nucleic acid concentration was determined by measuring $A_{260}$ (NanoDrop 2000; Thermo Fisher Scientific), and 100 ng DNA was used per PCR (final volume 30 $\mu$l). The PCR products were analyzed by electrophoresis on a 2.5% agarose gel. For mutation identification, PCR amplicons were purified using the Wizard SV Gel and PCR Clean-Up System kits (Promega), followed by Sanger sequencing (Eurofins). The DNA sequences were compared with the wild-type *cln3* sequence (ENSDART00000055170) listed in the ENSEMBL database. Primer sequences are given in Table S6.

### cDNA sequencing

Total RNA extraction and cDNA synthesis were performed as described previously (35). 1 $\mu$l cDNA of WT and CRISPR *cln3* mutant lines was used as a template to amplify the sequence spanning exon 2 to exon 7 (containing the *cln3* sgRNA target site) of the cln3 transcript, and the amplicons were cloned into the pGEMT vector following the manufacturer's instructions (Promega). Selected positive clones were sent for sequencing. To sequence full-length WT and mutant *cln3* transcripts, *cln3* was amplified from MUT1 and MUT2 cDNA and cloned into pDONR221 using the Gateway technology (61). Primer sequences used for the cloning experiments are given in Table S6.

## Locomotor behavioral assays

Locomotor activity of 5 dpf larvae in response to light–dark conditions or pro-convulsion treatment was analyzed using a DanioVision system equipped with a temperature control unit (Noldus Information Technology). The EthoVision XT 11.5 software (Noldus Information Technology) was used to monitor the movements of individually swimming larvae. Motion values were expressed as mean swimming velocity (mm/s). All tracking experiments were performed at least as independent triplicates (with at least 8 larvae per genotype per replicate). One representative experiment was selected for the preparation of the figures. Technical outliers (i.e., no detection of the larva by the system) were removed from the quantification. Statistical analysis was performed using GraphPad Prism analysis software (version 8.2.0).

### Dark–light–dark and global activity tracking

One hour before the recording, 5 dpf zebrafish larvae were carefully transferred into each well of a 24-well plate, with 1 ml Danieau's medium per well. The plate was then incubated at 28°C in the dark. For activity recording, each plate was placed in the DanioVision system and the larvae were allowed to habituate in the dark to the new environment for 15 min. After this period, locomotor activity was continuously tracked using a dark–light–dark protocol (15-min intervals). For the global activity test, larvae were placed in the DanioVision system to habituate in the tested lightning condition for 15 min, followed by locomotor activity recording for 45 min. All tested genotypes were represented by the same number of larvae in each 24-well plate to avoid any variations because of differences in experiment timing and handling. All experiments were performed during the day between 2 and 6 pm.

### Pro-convulsant treatment

For each experimental group, twelve 5 dpf zebrafish larvae were placed individually in 96-well plates. Each well contained 100 $\mu$l of Danieau's medium. Subsequently, 100 $\mu$l of pro-convulsant working solution (PTX or PTZ) was added to each well to obtain final concentration ranges of 0.01–1 mM for PTX and 5–20 mM for PTZ. Immediately after addition of the drug or 10 min later, locomotor behavior was recorded in the DanioVision system for 1 h in the dark.

## RNA extraction and qPCR

Adult zebrafish were euthanized in buffered 0.04% MS222 before organ dissection, and 30 larvae at the indicated developmental stages were euthanized by hypothermia. For total RNA extraction, organs and larvae were placed in Precellys tubes containing 1.4-mm ceramic beads and 900 $\mu$l TRIzol (Thermo Fisher Scientific). Samples were homogenized for 30 s at 6,000 rpm (Precellys 24, Bertin Instruments) and at 4°C, and 200 $\mu$l chloroform was added to the lysates, followed by centrifugation for 15 min at 15,700$g$ and 4°C. The upper aqueous phase was transferred into a fresh Eppendorf tube containing 450 $\mu$l isopropanol, followed by vortexing. After a 10-min incubation at room temperature, samples were centrifuged again for 15 min at 13,000$g$ and 4°C. Pellets were washed with 1 ml 70% ethanol and dried for 5–10 min. Finally, pellets were resuspended in 30 $\mu$l Milli-Q water, the RNA concentration was determined by measuring $A_{260}$ (NanoDrop 2000), and the RNA quality was evaluated by agarose gel electrophoresis. After treatment with DNase I (RNase-free; Sigma-Aldrich), total RNA (2 $\mu$g per reaction) was reverse-transcribed with the SuperScript III reverse transcriptase (Invitrogen) according to the manufacturer's instructions. 10-fold diluted cDNA samples (2 $\mu$l) were added to a mixture (10 $\mu$l total volume) containing iQ SYBR Green Supermix (5 $\mu$l; Bio-Rad) and the primers of interest at a final concentration of 10 $\mu$M. Gene expression was analyzed by quantitative PCR (qPCR) in 384-well plates using the LightCycler 480 instrument (Roche). For each tissue, qPCR was performed for three biological replicates, with two technical replicates per biological replicate. Expression levels of *cln3* in the indicated tissues were calculated relative to the eye tissue using the $2^{-\Delta\Delta Ct}$ method (62). *Ef1α* and *rpl13α* were used as reference genes for normalization. All qPCR primer sequences are given in Table S6.

## Untargeted metabolomics analysis using HILIC-HRMS

Zebrafish larvae at 5 dpf were washed twice with 2 ml 0.3X Danieau's solution and transferred to labeled and pre-weighed Precellys tubes (40 larvae/tube). Larvae were euthanized on ice and stored

at −80°C. Before extraction, samples were lyophilized overnight, then weighed. Cold dichloromethane (DCM, 330 µl), methanol (170 µl), and 25 ceramic beads (φ 1.4 mm) were added to each tube, and samples were homogenized twice (3,000 rpm for 30 s) at −10°C using a Precellys instrument (Bertin Instruments). After vortexing, homogenates (400 µl) were transferred to 2-ml Eppendorf tubes containing 117 µl of cooled Milli-Q water, followed by mixing through inversion. In a thermomixer, samples were further mixed for 20 min at 1,000 rpm and 4°C. Finally, samples were centrifuged at maximum speed (16,000$g$) for 20 min at 4°C, and 100 µl of the upper aqueous layer was collected for metabolomics analysis, whereas 100 µl of the organic lower phase was collected for lipidomics analysis. The non-polar extracts were vacuum-concentrated until dryness (CentriVap Benchtop Vacuum Concentrator; LABCONCO).

Hydrophilic interaction liquid chromatography (HILIC) was carried out using a SeQuant ZIC-pHILIC column (5 µm, 2.1 × 150 mm) equipped with a guard column of the same chemistry (5 µm, 2.1 × 20 mm). The flow rate was set at 0.20 ml/min using aqueous 20 mM ammonium acetate, pH 9.0 (A), and 20 mM ammonium acetate, pH 9.0, in acetonitrile (B) as mobile phases. The mobile phase gradient started at 90% B for 1.5 min before linearly ramping to 20% B over 14.5 min. This condition was maintained for 2 min before returning to the starting mobile phase conditions over 2 min. The column was allowed to re-equilibrate for 10 min at 90% B before the next injection. Mass spectrometry was performed separately in a positive and negative mode using a High Field Q Exactive mass spectrometer (Thermo Fisher Scientific). MS1 data acquisition settings were as follows: mass resolution of 120,000, default charge of 1, and automatic gain control of $1 \times 10^6$ with a maximum injection time (IT) of 70 ms over a scan range of 60–900 $m/z$. For MS/MS acquisition, MS/MS events were triggered in a data-dependent dynamic exclusion fashion where the top 5 highest intensity ions were fragmented with a dynamic exclusion window of 10 s. The automatic gain control was set to $8 \times 10^3$ with a maximum IT of 70 ms. The minimum ion intensity was $1 \times 10^5$ with an isolation window of 1 Da and HCD collision energy of 30 NCE (nominal collision energy).

Untargeted data analysis was performed using Progenesis QI (version 2.2; Waters) for the two different ionization modes separately. For peak picking, a minimum peak width of 0.1 min was used. Three consecutive metabolite annotation searches were performed with Progenesis software. The first round used the provided METLIN MS/MS database, whereas the second round used the following subset of selected databases for MS1 and (in silico) MS2 annotations within the ChemSpider mode: BioCyc, Human Metabolome Database, Kyoto Encyclopedia of Genes and Genomes, LipidMaps, PubMed, Peptides, and Yeast Metabolome Database. In the last round, a LipidBlast search was performed (63). Only proton loss ([M−H]⁻) and proton gain ([M+H]⁺) were considered during peak annotation for negative and positive modes, respectively. The mass error for MS1 and MS2 searches was both set at 5 ppm. For the remaining Progenesis settings, the default setting was selected. Peak area normalization was performed using the "normalize to all compounds" method provided by Progenesis software. Quantitative extracted ion chromatogram areas together with the primary metabolite annotations for both the batches are available in Tables S1 and S2, respectively. The annotation with the highest Progenesis score was chosen as the primary annotation. In case of identical

Progenesis scores, the alphabetically first annotation was designated as the primary annotation. All (redundant) compound annotations for both batches together with Progenesis scores and MS2 identification scores (when available) are given in Table S7 together with ChemSpider or METLIN identifiers. Selected metabolites of interest were validated with standard compounds (Table S3).

Univariate and multivariate statistical analyses were performed using the statistical program R (ver. 4.01). For univariate analysis, the extracted ion chromatogram areas for the annotated features were log₁₀-transformed and $t$-tested using the packages *Class-Comparison* (ver 3.1.8) and *stats* (ver 4.0.2). For multivariate analysis, principal component analysis was performed using the *FactoMineR* package, ver 2.3 (64). Metabolite enrichment analysis was performed using the *MetaboAnalystR* R package (ver 3.0.2) (65) including statistically significant differential features that had a primary annotation that could be mapped to a Human Metabolome Database identifier.

**Targeted metabolomics analyses using LC-MS**

The homogenate was incubated at 4°C with shaking (1,400 rpm) for 10 min (Eppendorf Thermomixer), followed by centrifugation at 21,000$g$ for 10 min at 4°C. Then, 70 µl of the supernatant was filtered and transferred into LC vials with microinsert for amino acid and glycerophosphodiester analysis. In addition, 100 µl of the supernatant was dried in a rotary vacuum concentrator at −4°C. Dried extracts were reconstituted in 100 µl water/acetonitrile (1:1, vol/vol), filtered, and submitted to carnitine analysis. The deuterated amino acid and carnitine mixture (MassChrom Internal Standard 5700457004), used as "Internal Standard Mix," was reconstituted in 80% methanol and further diluted according to the MassChrom Internal Standard 57004 manual. Extraction fluid was prepared by an additional 1:10 dilution in 80% methanol in water.

For metabolite extraction, 1 ml extraction fluid was added to 40 or 10 zebrafish larvae and homogenized using a Precellys Evolution homogenizer (Bertin Technologies) in the presence of 500 mg ceramic beads (1.4 mm) for 30 s at 6,000 rpm and 0 to 5°C. The homogenate was incubated at 4°C with shaking (1,400 rpm) for 10 min (Eppendorf Thermomixer), followed by centrifugation at 21,000$g$ for 10 min at 4°C. Then, 70 µl of the supernatant was filtered and transferred into LC vials with microinsert for amino acid and glycerophosphodiester analysis. In addition, 100 µl of the supernatant was dried in a rotary vacuum concentrator at −4°C. Dried extracts were reconstituted in 100 µl water/acetonitrile (1:1, vol/vol), filtered, and submitted to carnitine analysis.

For amino acid and glycerophosphodiester analysis, measurements were performed using a UHPLC Vanquish system coupled to an Orbitrap Exploris 240 mass spectrometer (Thermo Fisher Scientific) equipped with an OptaMax NG electrospray ionization source. Chromatography was carried out with a SeQuant ZIC-pHILIC 5-µm polymer column (150 × 2.1 mm) protected by a SeQuant ZIC-pHILIC Guard pre-column (20 × 2.1 mm). The column temperature was maintained at 45°C. Mobile phases consisted of 20 mmol/liter ammonium carbonate in water, pH 9.2, containing 0.1% InfinityLab deactivator additive (eluent A) and 100% acetonitrile (eluent B). The gradient was as follows: 0 min, 80% B; 3 min, 80% B; 18 min, 20% B;

19 min, 20% B; 20 min, 80% B; and 30 min, 80% B. The flow rate was set to 0.2 and 0.4 ml/min for the re-equilibration phase. The injection volume was 5 μl. MS analysis was performed using electrospray ionization with polarity switching enabled (+ESI/−ESI). The source parameters were applied as follows: sheath gas flow rate, 35; aux gas flow rate, 7; sweep gas flow rate, 0; RF lens, 70%; ion transfer tube temperature, 320°C; and vaporizer temperature, 275°C. Spray voltage was set to 3,000 V in a positive and −3,000 V in a negative mode. The Orbitrap mass analyzer was operated at a resolving power of 60,000 in a full-scan mode (scan range: m/z 75–1,000; standard automatic gain control; and maximum injection time: 100 ms). Data were acquired with Thermo Xcalibur software (version 4.5.474.0) and analyzed with TraceFinder (version 5.1). Target compounds were identified by retention time, accurate mass, and isotopic pattern based on authentic standards. In addition, the identity of all targets was confirmed by accurate mass, MS/MS experiments, and standard addition experiments, when authentic standards were available. Briefly, annotation of the compounds of interest was achieved by isolating the precursor ion and fragmenting it into the product ions. The resulting product ion spectrum was used for library matching and qualitative MS interpretation. The procedural error was corrected using the response ratio of the integrated peak area of the target compound to the integrated peak area of the dedicated internal standard. The synthesis of the GPI and GPE standards was performed as described in Laqtom et al (21).

For carnitine analysis, measurements were performed using ExionLC coupled to a 7500 Triple quad MS (SCIEX) equipped with an OptiFlow Pro ion source. The ion source was operated in an electrospray ionization mode. Chromatography was accomplished using a Waters ACQUITY UPLC CSH C18 1.7-μm column (2.1 × 100 mm) protected by a VanGuard pre-column (2.1 × 5 mm). The column temperature was maintained at 40°C. The mobile phases consisted of 10 mmol/liter ammonium formate in water, pH unadjusted (eluent A), and acetonitrile containing 0.1% formic acid (eluent B). The flow rate was set to 0.3 ml/min. The LC method consisted of 2-min isocratic delivery of 5% B, a 10-min linear gradient to 95% B, and a 3-min isocratic delivery of 95% B, followed by a re-equilibration phase on starting conditions at 5% B for 5 min. The injection volume was 5 μl. Target compounds were measured in a scheduled multiple reaction monitoring mode. Specific transitions of each target analyte are given in Table S8. The source and gas parameters applied were as follows: ion source gases 1 and 2 were maintained at 40 and 70 ψ, respectively; curtain gas was maintained at 40 ψ; CAD gas was maintained at 10; and source temperature was held at 550°C. Spray voltage was set to 2,000 V in a positive ion mode. Mass spectrometric data were acquired with SCIEX OS (version 3.0.0) and analyzed with MultiQuant (version 3.0.3). Target compounds were identified by retention time and ion ratio. In addition, the identity of all targets was confirmed by MS/MS. The data set was normalized using the response ratio of the integrated peak area of target compounds to the integrated peak area of the dedicated internal standard. D2-citrulline from the standard mix was used for normalization of the GPDs.

### Untargeted lipidomics analysis using LC-HRMS

The non-polar extracts were reconstituted in 100 μl methanol: toluene (9:1). Untargeted lipidomics analysis was adapted from

reference (66). Briefly, lipid separation was accomplished by chromatography on an ACQUITY UPLC CSH C18 column (1.7 μm, 100 × 2.1 mm) equipped with an ACQUITY UPLC CSH C18 VanGuard pre-column (1.7 μm, 5 × 2.1 mm) (Waters) using the same LC-HRMS instrument as for untargeted metabolomics analysis. The column temperature was set at 65°C and the mobile phase flow rate at 0.6 ml/min. Mobile phases (A) 60:40 (vol/vol) acetonitrile:water with ammonium formate (10 mM) and formic acid (0.1%) and (B) 90:10 (vol/vol) isopropanol:acetonitrile with ammonium formate (10 mM) and formic acid (0.1%) were mixed according to the following gradient program: 0 min, 15% (B); 0–2 min, 30% (B); 2–2.5 min, 48% (B); 2.5–11 min, 82% (B); 11–11.5 min, 99% (B); 11.5–12 min, 99% (B); 12–12.1 min, 15% (B); and 12.1–15 min, 15% (B). The sample temperature was maintained at 4°C. Peak picking, lipid MS2 annotation, and data normalization by the LOESS algorithm were performed following the protocol for lipid analysis using MS-DIAL (version 4.48) (67, 68).

### Targeted lipidomics analysis using LC-MS

Lipid extraction and targeted lipidomics analysis were performed by Lipometrix at the KU Leuven. An amount of cells containing 10 μg of DNA was homogenized in 700 μl of water with a handheld sonicator and was mixed with 800 μl HCl(1 M):CH3OH 1:8 (vol/vol), 900 μl CHCl3, 200 μg/ml of the antioxidant 2,6-di-tert-butyl-4-methylphenol (BHT; Sigma-Aldrich), 3 μl of UltimateSPLASH ONE internal standard mix (#330820; Avanti Polar Lipids), and 3 μl of a 10 μg/ml solution of oleoyl-L-carnitine-d3 (#26578; Cayman Chem). After vortexing and centrifugation, the lower organic fraction was collected and evaporated using a Savant SpeedVac spd111v centrifuge (Thermo Fisher Scientific) at room temperature and the remaining lipid pellet was stored at −20°C under argon.

Just before mass spectrometry analysis, lipid pellets were reconstituted in 100% ethanol. Lipid species were analyzed by liquid chromatography/electrospray ionization tandem mass spectrometry (LC-ESI/MS/MS) on a Nexera X2 UHPLC system (Shimadzu) coupled with hybrid triple quadrupole/linear ion trap mass spectrometer (6,500+ QTRAP System; AB SCIEX). For acylcarnitine, PG, and BMP measurement, chromatographic separation was performed on a Luna NH₂ column (100 × 2 mm, 3 μm; Phenomenex) maintained at 35°C using mobile phase A (2 mM ammonium acetate in dichloromethane/acetonitrile 7:93 [vol/vol]) and mobile phase B (2 mM ammonium acetate in water/acetonitrile 50:50 [vol/vol]) in the following gradient: 0–2 min, 0% B; 2–11 min, 0% B → 50% B; 11–12.5 min, 50% B → 100% B; 12.5–15 min, 100% B; 15–15.1 min, 100% B → 0% B; and 15.1–17 min, 0% B, at a flow rate of 0.2 ml/min, which was increased to 0.7 ml/min from 2 to 15 min. For all other lipids, chromatographic separation was performed on an XBridge Amide column (150 × 4.6 mm, 3.5 μm; Waters) maintained at 35°C using mobile phase A (1 mM ammonium acetate in water/acetonitrile 5:95 [vol/vol]) and mobile phase B (1 mM ammonium acetate in water/acetonitrile 50:50 [vol/vol]) in the following gradient: 0–6 min, 0% B → 6% B; 6–10 min, 6% B → 25% B; 10–11 min, 25% B → 98% B; 11–13 min, 98% B → 100% B; 13–19 min, 100% B; and 19–24 min, 0% B, at a flow rate of 0.7 ml/min, which was increased to 1.5 ml/min from 13 min onward. Sphingomyelins (SM), cholesterol esters (CE), ceramides (CER), dihydroceramides (DCER), hexosylceramides (HCER),

lactosylceramides (LCER), and acylcarnitines were measured in a positive ion mode with a production ion of 184.1, 369.4, 264.4, 266.4, 264.4, 264.4, and 85.0, respectively. Triacylglycerides (TAG) and diacylglycerides (DAG) were measured in a positive ion mode with a neutral loss for one of the fatty acyl moieties. Phosphatidylcholine (PC), lysophosphatidylcholine (LPC), phosphatidylethanolamine (PE), lyso-phosphatidylethanolamine, phosphatidylglycerol (PG), BMP, phosphatidylinositol (PI), and phosphatidylserine (PS) were measured in a negative ion mode by fatty acyl fragment ions. Lipid quantification was performed by scheduled multiple reaction monitoring, the transitions being based on the neutral losses or the typical product ions as described above. The instrument parameters were as follows: curtain gas, 35 $\psi$; collision gas, 8 a.u. (medium); IonSpray voltage, 5,500 V and −4,500 V; temperature, 550°C; ion source gas 1, 50 $\psi$; ion source gas 2, 60 $\psi$; declustering potential, 60 V and −80 V; entrance potential, 10 V and −10 V; and collision cell exit potential, 15 V and −15 V. The following fatty acyl moieties were taken into account for the lipidomics analysis: 14:0, 14:1, 16:0, 16:1, 16:2, 18:0, 18:1, 18:2, 18:3, 20:0, 20:1, 20:2, 20:3, 20:4, 20:5, 22:0, 22:1, 22:2, 22:4, 22:5, and 22:6 except for TGs that considered 16:0, 16:1, 18:0, 18:1, 18:2, 18:3, 20:3, 20:4, 20:5, 22:2, 22:3, 22:4, 22:5, and 22:6.

Peak integration was performed with MultiQuant software, version 3.0.3. Lipid species signals were corrected for isotopic contributions (calculated with Python Molmass 2019.1.1) and were quantified based on internal standard signals, in adherence to the guidelines of the Lipidomics Standards Initiative (level 2–type quantification as defined by the LSI).

### Cathepsin D activity assay

Cathepsin D activity in zebrafish extracts was assayed using a protocol adapted from references 56 and 69. Total protein was extracted from a pool of 60 WT and mutant (MUT1) larvae at 5 dpf. Samples were homogenized at 4°C in 300 $\mu$l lysis buffer (0.1 M Na acetate, 1 mM EDTA, and 0.1% Triton X-100, pH 4.0) for 60 s at 6,000 rpm using a Precellys instrument (Bertin Instruments). The homogenates were centrifuged at 4°C for 30 min at 9,300$g$, and the total protein concentration in the supernatants was determined using Pierce BCA Protein Assay Kit (Thermo Fisher Scientific).

Cathepsin D activity was assayed in a reaction mixture (175 $\mu$l total volume) containing 28.6 mM sodium acetate buffer (pH 4.0), 1.4% (wt/vol) hemoglobin, and 20 $\mu$g total protein extract. Mixtures were incubated at 37°C for 3 h, and reactions were stopped by addition of 150 $\mu$l of 15% (wt/vol) trichloroacetic acid (TCA) followed by centrifugation at 16,000$g$ for 6 min at room temperature. Supernatants (200 $\mu$l) were neutralized by addition of 16 $\mu$l of 4 M NaOH. The TCA-soluble peptides were measured using Pierce BCA Protein Assay Kit.

### Cerebral organoid derivation

Cerebral organoids were the same as the ones used in Gomez-Giro et al (18). They were derived from isogenic control and CLN3$^{Q352X}$ iPSCs, following the protocol described by Lancaster et al (36) and extensively characterized in the study described previously (18). Cerebral organoids were maintained for 55 d after the beginning of

the differentiation (the total duration in culture was 66 d). At the time of collection, organoids were washed once with PBS, snap-frozen, and stored at −80°C until they were processed. For each genotype, two organoids of four independent organoid derivations were pooled for the metabolomics analysis.

### Electron microscopy methods

Zebrafish larvae at 6 dpf were fixed in 4% glutaraldehyde (ems), 1% paraformaldehyde (ems), and 2.5 mM CaCl$_2$ in 0.1 M cacodylate buffer (Sigma-Aldrich) for 1.5 h, and post-fixed in 0.1% O × O$_4$ for 1 h. After dehydration in increasing concentrations of ethanol, the larvae were embedded in Agar 100 resin (Agar Scientific) and cut into 60-nm sagittal sections in two planes, paramedial and at mid-eye level on an ultramicrotome (RMC Boeckeler). Sections were mounted to 2 × 1 mm copper slot grids (gilder grids) and imaged on a Zeiss Sigma 300 SEM with a STEM detector at an acceleration voltage of 30 kV. For reference of the zebrafish anatomy, the ZFIN zebrafish atlas by Diever et al was used (https://zfin.org/zf_info/anatomy.html).

## Data Availability

All relevant data have been included in the supporting information. This study includes no data deposited in external repositories.

## Supplementary Information

## Acknowledgements

We are grateful to the fish caretaker team at the LCSB Aquatic Platform for their valuable daily work. We would also like to thank the LCSB Metabolomics Platform for providing technical and analytical support. We acknowledge the use of Biorender.com for Fig 4 (panels A and B) and Fig 8. This work was supported in part by donations from the ATOZ Foundation and Mr. Norbert Becker to CL Linster and by a Pélican award from the Fondation du Pélican de Mie et Pierre Hippert-Faber to U Heins-Marroquin. CL Linster, RR Singh, and EL Schymanski acknowledge funding support from the Luxembourg National Research Fund (FNR) for projects C20/BM/14701042 (CL Linster) and A18/BM/12341006 (RR Singh and EL Schymanski).

### Author Contributions

U Heins-Marroquin: conceptualization, formal analysis, validation, investigation, visualization, and writing—original draft, review, and editing.

RR Singh: formal analysis, investigation, methodology, and writing—original draft.

S Perathoner: investigation, methodology, and writing—review and editing.

F Gavotto: methodology and writing—review and editing.

C Merino Ruiz: investigation and writing—review and editing.

M Patraskaki: methodology and writing—review and editing.

G Gomez-Giro: investigation and writing—review and editing.

F Kleine Borgmann: investigation, methodology, and writing—review and editing.

M Meyer: methodology and writing—review and editing.

A Carpentier: methodology and writing—review and editing.

MO Warmoes: data curation, formal analysis, visualization, and writing—original draft.

C Jäger: validation, investigation, methodology, and writing—review and editing.

M Mittelbronn: supervision and writing—review and editing.

JC Schwamborn: supervision and writing—review and editing.

ML Cordero-Maldonado: investigation and writing—original draft.

AD Crawford: conceptualization, supervision, and writing—original draft.

EL Schymanski: supervision and writing—original draft.

CL Linster: conceptualization, supervision, funding acquisition, investigation, project administration, and writing—original draft, review, and editing.

## Conflict of Interest Statement

The authors declare that they have no conflict of interest.

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
