## [Reviewer comments · Life Science Alliance]

CLN3 deficiency leads to neurological and metabolic perturbations during early development

Ursula Heins-Marroquin, Randolph Singh, Simon Perathoner, Floriane Gavotto, Carla Merino Ruiz, Myrto Patraskaki, Gemma Gomez-Giro, Felix Kleine Borgmann, Melanie Meyer, Anaïs Carpentier, Marc Warmoes, Christian Jaeger, Michel Mittelbronn, Jens Schwamborn, Maria Lorena Cordero-Maldonado, Alexander Crawford, Emma Schymanski, and Carole Linster
DOI: <https://doi.org/10.26508/lsa.202302057>

Corresponding author(s): Carole Linster, University of Luxembourg

Review Timeline:

Submission Date:	2023-03-24
Editorial Decision:	2023-04-20
Revision Received:	2023-11-15
Editorial Decision:	2023-12-11
Revision Received:	2023-12-13
Accepted:	2023-12-14

Transaction Report:

April 20, 2023

Re: Life Science Alliance manuscript #LSA-2023-02057-T

Prof. Carole Liette Linster
University of Luxembourg
Luxembourg Centre for Systems Biomedicine
7, Ave des Hauts-Fourneaux
Esch-sur-Alzette 4362
Luxembourg

Dear Dr. Linster,

Thank you for submitting your manuscript entitled "CLN3 deficiency leads to neurological and metabolic perturbations during early development" to Life Science Alliance. The manuscript was assessed by expert reviewers, whose comments are appended to this letter. We invite you to submit a revised manuscript addressing the Reviewer comments.

Please note that it is not necessary to address Reviewer 3's major comments #2 and 3, unless the data is readily available.

Thank you for this interesting contribution to Life Science Alliance. We are looking forward to receiving your revised manuscript.

Sincerely,

B. MANUSCRIPT ORGANIZATION AND FORMATTING:

Reviewer #1 (Comments to the Authors (Required)):

Review Heins-Marroquin,
CLN3 deficiency leads to neurological and metabolic perturbations during early development.

The authors describe generation of first knockout zebrafish model for CLN3/Batten disease, a severe neurodegenerative disease. Previous work by Wager et al described a morpholino induced phenotype which seemed to resemble the human phenotypes. The study by Heinz-Marroquin sets of by an attempt to reproduce this morpholino-based study, without observing the strong phenotypes observed in the earlier study. Next, they generate two loss-of-function alleles for *cln3* in the zebrafish using *crispr-cas9*, also without observing strong early phenotypes, thus challenging Wager et al. Heinz-Marroquin et al show convincingly that loss of *cln3* in the zebrafish is not associated with a strong developmental phenotype.

- The discrepancies between the published morphant phenotypes and the morphant and genetic studies here can be discussed in stronger terms in my opinion, also referring to Stainier et al and following the reasoning in flow diagram in there (PMID:29049395). Please also discuss the maternal zygosity of the animals used in the different experiments in this manuscript. These guidelines and realization of the issues with morphant phenotypes are important for the zebrafish community. The current study shows an excellent example why morphant studies should be used with caution.

Next the authors studied the mutant lines for behavioral phenotypes and observed increased baseline activity, decreased light sensitivity and increased sensitivity to pro-convulsive drugs. They observe subtle but reproducible differences in light response and PTX response in both mutant lines.

- Regarding the light response the authors write 'although capable to react to light changes, may have an impaired visual acuity.' I do not see any evidence that the difference in light response is due to an issue in the eye rather than the brain. Please elaborate on this.

- For fig 3B, D F and G please indicate whether this is showing representative tracks of a single experiment with multiple larvae or an average over different experiments done on different days. Fig 3F: I guess the blue and black triangles in the legend should be exchanged?

Next the metabolic consequences of *cln3* mutations were studied using polar untargeted metabolomics and lipidomics. These studies revealed GPD and BMP accumulation in *cln3* larvae, confirming the relevance of the recently discovered biomarker molecules for *cln3*/Batten disease in an additional vertebrate model; moreover the metabolic aberrations are measurable at a very early stage.

- In the abstract GPD and BMP are mentioned as biomarkers, however the relevant studies highlighting the importance of the biomarkers are not mentioned in the introduction, please add those.
- The untargeted metabolomics data is divided in a discovery and a validation set; would a combination of both datasets yield a stronger discovery dataset? In my view the validation lies in the LC-MS validations and observations in other models. In line with this, LC-MS validation of metabolites in the *Mut2*-line would be a valuable addition to the manuscript.
- The manuscript would become stronger if LC-MS metabolic measurements in older animals are also included, as this would provide insights in the development of the metabolic phenotype.

Minor comment:

- Page 5: 'These observations only partially overlap with human and mouse expression data, in which CLN3 is expressed at similar levels in all the analyzed organs (Yue et al. 2014; Fagerberg et al. 2014). Our results suggest that, in zebrafish, *cln3* might play an important role more specifically in the brain, eyes, and gonads during adulthood.' Although already carefully phrased, I am not sure whether the observation of different relative mRNA expression across tissues warrants any conclusion about functionality.

Taken together this is a well written manuscript of great value for the research community working on lysosomal storage diseases and lysosome biology.

Reviewer #2 (Comments to the Authors (Required)):

This paper describes the generation of morphant models and CRISPR-generated CLN3 lines of Zebrafish which unexpectedly showed no obvious gross phenotypes if a knockout and a human cerebral organoid model modelling a specific mutation in CLN3. The most striking finding was that in-depth metabolomics and lipidomics analyses revealed significant accumulation of several glycerophosphodiester (GPDs), especially GPI and a global decrease of bis(monoacylglycero)phosphate (BMP) species. This supports their utility as potential biomarkers.

The paper is well written and contains a wealth of well presented data throughout. The independent discovery of depletion of BMPs and raised GPDs is pleasing as this is consistent with both old and recent publications, respectively. (It is not necessary to say that the work on GDPs was done in parallel more than once).

The main concern is whether the strains are true knockouts, as this is not proven.

Comments on specific sections and suggestions follow:

Introduction:

Add that most common mutation in juvenile CLN3 disease is 1 kb deletion (actually 966b deletion, so correct any reference to 1.02kb as in Fig2E) as this is relevant to geno-phenotype correlations of disease models, and also if the normal disease is not actually a disease from loss of function (see Kitzmuller et al 2008 which describes at least two transcripts for CLN3 1 kb deletion in human cells, and Minnis et al 2021 which models one of these in yeast) - this is picked up in the discussion for Kitzmuller et al.

Results:

The chimeric P0 fish have pigment changes due to targeting *slc45a2* as well as *cln3*, but this is not mentioned in the results, nor how it was ensured that the final *cln3* mutants strains did not contain *slc45a2* mutations. Did they ensure this?

Are all of the het fish (fig 2F top fish) smaller than the mutants? Was fish length quantified? Should this be included? Should this be included for adults?

Were previously used MO sequenced tried for comparison? As the phenotypes are so different to that reported in Wagner et al.

How can the authors be sure the strains are true knockouts without using long read RNA or alternative primers in different regions, given that they did not use antibodies to test the presence or absence of CLN3 protein? Can they sequence check the *Cln3* locus? I strongly suggest using alternative names in case these turn out not to be true knockouts with absolutely no *Cln3* produced.

For both e2i2-MO and i2e3-MO, there is a disruption within the primer targeting sequence and a shift is observed. However, the sequence of these PCR products are not shown to assess if the predicted Stop codon is actually introduced.

Was exon skipping considered which is highly likely and could produce partially function *cln3* in all mutants including the CRISPRs? As in Kitzmuller et al and other CLN3 papers.

The authors made *slc45a2* mutants in the same fish that they were making *cln3* mutants. I am concerned that there is no mention of ensuring that the *cln3* mutant lines did not contain the *slc45a2* mutation (even as heterozygotes). If it did, this would hamper interpretation of *cln3* mutant phenotypes.

The authors have not followed ARRIVE guidelines as they do not state how many animals were used per experiment.

p.7 Correct reference to 1.02kb - it is 1 kb deletion and there are at least two transcripts for this disease allele Kitzmuller et al 2008. Did you look for other transcripts arising from exon skipping??

P12. There is no reference for reduced CTSD activity in CLN3 Batten disease, which should be added, or instead as CLN10 disease is caused by mutations in CTSD then this is good reason to look.

They should provide the transition tables for the targeted metabolite assays (GPDs most importantly) as they have only done this for the carnitines and AA assays. This is quite important for other groups to replicate/compare future findings.

They should clarify the internal standards they have used for the targeted analysis - it is unclear what was used for the GPDs (IS has been used before).

Discussion:

FigS1A- add a comment that the alignment is agreement with Wager et al, and extends the analysis to further species.
Discuss RNA expression in FigS1B in relation to in situ hybridisation in Wager et al.

If the different background effect is not responsible for the difference in the Wager et al morphant studies, what could the reason be? Is it worth adding a sentence to say this is unexplained as it appears that the majority of transcripts are affected? Could the amount of reduction of normal transcripts in relation to the heterozygous state (which is phenotypically normal) be discussed, and compared to previous papers estimating how much CLN3 protein or transcript is required for normal development and absence of phenotypes.

p.16 'tissular'??

Figures:

2E Correct reference to 1.02kb - it is 1 kb deletion and there are at least two transcripts for this disease allele Kitzmuller et al 2008

2F The first fish is smaller than the others. Is this picture at the same magnification?

S2 shows 11 hydrophobic regions but Fig 2E shows only six. Reconcile?

S1A show exact species in legend and ideally also the figure. Clarify if point mutations (ie single nucleotide changes) or missense mutations affecting a single amino acid. Indicate 1 kb deletion. There are other potential posttranslational sites, so either show all or none

S2 shows 11 hydrophobic regions but Fig 2E shows only six. Reconcile?

It would be better to report normalised abundance/ 40 larvae in the graph axis instead of mentioning in the legend.

S5A a 3-D PCA on a 2D page needs the 3rd vector direction indicated. Perhaps present as a biplot as most of the separation is in the PC1 (25.7%).

Methods:

It would be useful to indicate introns versus UTR and coding sequences in the MO sequences e.g. by using upper and lower case.

What strain of fish was injected to generate P0 fish? Were CRISPR targets sequenced in the strain that were injected? Were F1 and F2s checked for lack of slc45a2 mutations? If so, how?

It would be useful to state how the mutants and heterozygotes were generated e.g. by incrossing heterozygotes, so they were siblings raised together, or were some homozygous mutants generated by incrossing homozygous adults, or homozygous x heterozygous adults. Were all the mutants zygotic, or were some maternal zygotic?

Reviewer #3 (Comments to the Authors (Required)):

Zebrafish have been successfully used to model a range of neurological and metabolic perturbations in different lysosomal storage diseases (Zhang, et al 2020). Heins-Marroquin and colleagues aimed at developing and characterizing a zebrafish model of Cln3 disease at the early developmental stages of life. Heins- Marroquin et al mutant models exhibit milder behavioral abnormalities than another published line (Wager, et al 2016). The mutant lines also recapitulate the major metabolic changes that were previously observed in human patients and some other mammalian cln3-deficient systems mainly, the abnormal levels of GPDs and BMPs.

Major comments to improve the manuscript:

I don't think the study is particularly novel and no notable technological advance acquired as much as I can see. The work has been done on the whole larvae level and no work at the subcellular level, though Cln3 is a lysosomal protein.

1. The authors could visualize (image) the lysosomes (morphology, size, count, volume) in WT or Het versus MUT
2. Besides the whole level, the author could use lysosome enrichment kit for lysosomal purification from tissues, and subsequent metabolic analyses. In particular, the major changing metabolites (GPDs) are trapped within the lysosome. Lysosomes represent a small fraction of the whole cell, therefore, the lysosomal changes will be diluted at the whole cell level. The enrichment for lysosomes will improve the results and add more value for using this zebrafish model to monitor the disease. This part has been discussed in Laqtom et al, 2022.
3. The authors generated a comprehensive list of dysregulated polar and non-polar metabolites, I would encourage the authors to follow up on novel results and reproduce them in their organoid model system.
4. I think this manuscript is long and can be written concisely.

Minor comments:

Sup Fig. 2B, are both reference genes stably expressed in all times of developmental stages? Show statistical significance in the figure.

In Discussion:

In the *Cln3* knockout mouse model, tissular GPD accumulation became evident only in samples enriched in lysosomes (lysoIP approach) and was shown for approximately 7-months old animals (Laqtom et al. 2022).

I think 'only' is inaccurate here. By looking at their Fig.3, there were statistically increased levels of GPDs in the whole tissue and cell, but it becomes much more evident in lysosomal-enriched samples.

Point by point response to Reviewers' comments

Reviewer #1 (Comments to the Authors (Required)):

Review Heins-Marroquin, CLN3 deficiency leads to neurological and metabolic perturbations during early development.

The authors describe generation of first knockout zebrafish model for CLN3/Batten disease, a severe neurodegenerative disease. Previous work by Wager et al described a morpholino induced phenotype which seemed to resemble the human phenotypes. The study by Heinz-Marroquin sets of by an attempt to reproduce this morpholino-based study, without observing the strong phenotypes observed in the earlier study. Next, they generate two loss-of-function alleles for *cln3* in the zebrafish using crispr-cas9, also without observing strong early phenotypes, thus challenging Wager et al. Heinz-Marroquin et al show convincingly that loss of *cln3* in the zebrafish is not associated with a strong developmental phenotype.

- The discrepancies between the published morphant phenotypes and the morphant and genetic studies here **can be discussed in stronger terms in my opinion**, also referring to Stainier et al and following the reasoning in flow diagram in there (PMID:29049395). Please also **discuss the maternal zygosity of the animals used in the different experiments in this manuscript**. These guidelines and realization of the issues with morphant phenotypes are important for the zebrafish community. The current study shows an excellent example why morphant studies should be used with caution.

We understand the relevance of the discrepancy between both studies, and we have expanded the discussion on this point (page 16, lines 454-471). We have also provided all relevant information concerning the maternal zygosity of the animals used in this study (File S4, Fig S2A).

Next the authors studied the mutant lines for behavioral phenotypes and observed increased baseline activity, decreased light sensitivity and increased sensitivity to pro-convulsive drugs. They observe subtle but reproducible differences in light response and PTX response in both mutant lines.

- Regarding the light response the authors write 'although capable to react to light changes, may have an **impaired visual acuity**.' I do not see any evidence that the difference in light response is due to an issue in the eye rather than the brain. Please elaborate on this. We agree with the reviewer that we have no direct evidence that the mutant larvae have a visual impairment, but our observations also do not allow us to exclude this possibility. The absence, in the mutant larvae, of the freeze startle response that can be seen (as would be expected) in the control larvae upon the dark to light switch (Fig. 3D) could be due to a neurological problem, but it may also be that the mutants fail to initially "see" this change in dark/light due to decreased visual acuity. We have now reformulated the sentence to reflect the reviewer's point of view (mutant phenotype could be due to a brain or eye issue) that we agree with (page 9, lines 246-247).

- For fig 3B, D F and G please indicate whether this is showing representative tracks of a single experiment with multiple larvae or an average over different experiments done on different days. Fig 3F: I guess the blue and black triangles in the legend should be exchanged?

Results in Figure 3B, D, F, and G are means of the indicated number of larvae obtained in one experiment, but they are representative of at least 3 independent experiments performed on different days (only one representative experiment is shown). We have revised the figure legend text to explain this more clearly. Thank you for catching our mistake in the legend of Figure 3F, we corrected it.

Next the metabolic consequences of *cln3* mutations were studied using polar untargeted metabolomics and lipidomics. These studies revealed GPD and BMP accumulation in *cln3* larvae, confirming the relevance of the recently discovered biomarker molecules for *cln3*/Batten disease in an additional vertebrate model; moreover the metabolic aberrations are measurable at a very early stage.

- In the abstract GPD and BMP are mentioned as biomarkers, however the relevant studies highlighting the importance of the biomarkers are not mentioned in the introduction, please add those.

Thank you very much for pointing this out, we have now included these references in the introduction (page 5, lines 126-129).

- The untargeted metabolomics data is divided in a discovery and a validation set; would a combination of both datasets yield a stronger discovery dataset? In my view the validation lies in the LC-MS validations and observations in other models. In line with this, LC-MS validation of metabolites in the *Mut2*-line would be a valuable addition to the manuscript.

During the preparation of the manuscript, we considered combining both sets. By doing so, we obtained a list of differential metabolites that largely overlaps with the lists obtained by analysing the two data sets separately, however, we observed a clear batch effect for both experiments as you can see in Figure 1 shown below. We could present only one data set in the paper, but we believe that presentation of the two independent experiments strengthens the results, by giving us the opportunity to filter for metabolites that were significantly changed in both data sets only.

We agree with the suggestion for the *MUT2* line and now performed metabolite analyses also in this line. We integrated the new results in Fig6A/D and FigS9 of the revised manuscript. As for *MUT1* larvae, the *MUT2* larvae show an accumulation of GPDs and a decrease of certain amino acids.

PCA analysis of the untargeted metabolomics data of batch 1 (WTb, KO b) and batch 2 (Wta, Koa) together.

Figure 6 Validation of glycerophosphodiester and amino acid changes by targeted LC-MS analyses. (A) Metabolite levels of glycerophosphodiester species (GPE, GPI, GPS, GPG, GPC) extracted from batches of 10 whole zebrafish larvae at 5 dpf (n = 40 larvae for each genotypes).

Figure 6 (D) Metabolite levels of amino acids differing significantly between WT and MUT1 zebrafish larvae at 5 dpf based on targeted LC-MS measurements. Data shown are means \pm SDs of eight biological

replicates for WT1/MUT1 (n = 200 larvae per genotype) and four biological replicates for WT2/MUT2 (n = 40 larvae per genotype).

Figure S9 (C) Amino acids that were not significantly altered in whole larvae extracts of WT and MUT, based on targeted analysis. Metabolite levels of amino acids differing significantly between WT and MUT1 zebrafish larvae at 5 dpf based on targeted LC-MS measurements. Data shown are means \pm SDs of eight biological replicates for WT1/MUT1 (n = 200 larvae per genotype) and four biological replicates for WT2/MUT2 (n = 40 larvae per genotype).

- The manuscript would become stronger if LC-MS metabolic measurements in older animals are also included, as this would provide insights in the development of the metabolic phenotype.

We fully agree with the comment and the metabolic measurements at different ages and organs are planned for a follow-up project. The main scope of this paper was the creation of CRISPR zebrafish mutants and their characterization at larval stages for the identification of the earliest phenotypes in the model. The measurement in older fish would have further delayed the submission of the work and we considered that it was important to show that, in contrast to what may be currently believed in the community, invalidation of *cln3* function is not lethal in zebrafish, but leads to a robust metabolic phenotype consistent with the one observed in higher vertebrate models and humans, which can be exploited to progress in our understanding of an apparently well conserved *Cln3* function. Of particular interest in the zebrafish model is the rapidity at which this phenotype develops.

Minor comment:

- Page 5: ‘These observations only partially overlap with human and mouse expression data, in which *CLN3* is expressed at similar levels in all the analyzed organs (Yue et al. 2014; Fagerberg et al. 2014). Our results suggest that, in zebrafish, *cln3* might play an important role more specifically in the brain, eyes, and gonads during adulthood.’ Although already carefully phrased, I am not sure whether the observation of different relative mRNA expression across tissues warrants any conclusion about functionality.

We fully agree that RNA expression levels give only a preliminary indication and cannot be used for drawing final conclusions concerning functionality. Unfortunately, we could not assess protein expression levels because of the lack of reliable antibodies that detect endogenous *Cln3* levels. We

revised the text to clarify this (page 6, lines 155-157) and further underlined that conclusions about functionality remain very speculative at this stage based on the mRNA results.

Taken together this is a well written manuscript of great value for the research community working on lysosomal storage diseases and lysosome biology.

We thank the reviewer for this positive conclusion.

Reviewer #2 (Comments to the Authors (Required)):

This paper describes the generation of morphant models and CRISPR-generated CLN3 lines of Zebrafish which unexpectedly showed no obvious gross phenotypes if a knockout and a human cerebral organoid model modelling a specific mutation in CLN3. The most striking finding was that in-depth metabolomics and lipidomics analyses revealed significant accumulation of several glycerophosphodiester (GPDs), especially GPI and a global decrease of bis(monoacylglycero)phosphate (BMP) species. This supports their utility as potential biomarkers.

The paper is well written and contains a wealth of well presented data throughout. The independent discovery of depletion of BMPs and raised GPDs is pleasing as this is consistent with both old and recent publications, respectively. (It is not necessary to say that the work on GPDs was done in parallel more than once).

We thank the reviewer for these positive comments and made sure to mention only once the parallel GPD work.

The main concern is whether the strains are true knockouts, as this is not proven.

Comments on specific sections and suggestions follow:

Introduction:

Add that most common mutation in juvenile CLN3 disease is 1 kb deletion (actually 966b deletion, so correct any reference to 1.02kb as in Fig2E) as this is relevant to geno-phenotype correlations of disease models, and also if the normal disease is not actually a disease from loss of function (see Kitzmuller et al 2008 which describes at least two transcripts for CLN3 1 kb deletion in human cells, and Minnis et al 2021 which models one of these in yeast) - this is picked up in the discussion for Kitzmuller et al.

We thank the reviewer for this suggestion and described the 1 kb deletion in the introduction (page 3, lines 65-69) of the revised manuscript (and also corrected 1.02 kb to 1 kb throughout). We also included the Minnis et al 2021 reference in the discussion where we mention the possibility of residual function for truncated forms of CLN3 (page 17, line 481-484).

Results:

The chimeric P0 fish have pigment changes due to targeting slc45a2 as well as cln3, but this is not mentioned in the results, nor how it was ensured that the final cln3 mutants strains did not contain slc45a2 mutations. Did they ensure this?

We targeted the albino (*alb*; *slc45a2*) locus in zebrafish as a convenient means to assess the efficiency of our CRISPR/Cas9 system. Loss-of-function of this protein leads to unpigmented melanophores in the body and in the eye, a phenotype that is readily visible in young larvae, as well as in adult zebrafish. Because of its obvious loss-of-function phenotype and its mendelian inheritance, the *slc45a2* locus is very often used as a proof-of-concept for zebrafish genome engineering (Irion et al., 2014, Vicencio et al., 2022, Moreno-Mateos et al., 2015, Jao et al., 2013). For all experiments shown in our study, larvae of the F3 and later generations (up to F12) were used. For the creation of the F3 generation, we incrossed *cln3* homozygous mutants of the F2 generation and, in parallel, pairs of WT siblings (or outcrossed *cln3* homozygous mutants with WT siblings if HETs were to be used as controls) to generate “isogenic” controls. All of our zebrafish generations (including F2) presented normal pigmentation, indicating that we have not accidentally selected an *slc45a2* mutation. Furthermore, all our experiments were performed in triplicate using different parental batches (for the same *cln3* mutation) and, in some instances, different generations to ensure that results are specific to *cln3* mutation and not to secondary mutations. Finally, the majority of results shown in the manuscript were obtained in two independent *cln3* deficient lines (MUT1 and MUT2). Considering all this, it seems unlikely that any of the phenotypes that we describe are due to mutations in the *slc45a2* gene. However, to firmly exclude the presence of an *slc45a2* mutation also at the heterozygous state, we sequenced the Crispr target site of the *slc45a2* gene in genomic DNA of F2 or F3 generation animals. The sequences obtained perfectly aligned with the wild-type sequence confirming the absence of *slc45a2* mutations in our lines. This verification is also now described in the revised manuscript (page 9 line, 223-226; File S3).

Are all of the het fish (fig 2F top fish) smaller than the mutants? Was fish length quantified? Should this be included? Should this be included for adults?

Thank you for the observation, we didn't realize the size difference between the larvae that we showed in Fig 2F. We now chose pictures of larvae with closely similar sizes for the revised figure to not mislead the readers. We quantified larval length and eye area and found no significant differences between MUT1 or MUT2 larvae and their respective wildtype controls. These results have been added to supplementary Figure 2 (panel F-G) in the revised manuscript.

Fig 2. (F) MUT1 and MUT2 mutants and heterozygous controls at 5dpf. Scale bar represents 500 μm . Homozygous mutants are morphologically indistinguishable from their heterozygous counterparts. WT, wildtype; zf, zebrafish; hu, human.

Fig. S2 (F-G) Body length and eye area of 6 dpf larvae (n=10 per genotype).

Were previously used MO sequenced tried for comparison? As the phenotypes are so different to that reported in Wagner et al.

No, we did not try the previously published morpholinos. In a previous work of our group, we encountered a similar situation in which a previous paper showed the expected disease phenotypes using morpholinos against the *atp13a2* gene in zebrafish. For that study, we ordered the exact same morpholinos than used in the published study and reproduced the published results (strong phenotype), however, we could not phenocopy the results using other morpholinos targeting the same gene or in the two *atp13a2* knockout lines that we also generated and analyzed for our study (Heins-Marroquin et al., 2019). Moreover, we injected the previously published morpholinos into our *atp13a2* mutant lines, which allowed us to confirm that embryonic lethality caused by the previously published MOs was the result of an off-target effect. Based on this prior experience and as it is known that 25% of morpholinos lead to off-target effects (Eisen and Smith, 2008), we decided in this *cln3* study to directly go for different morpholinos targeting similar sequences to confirm (or not) the previously published observations. The translation-blocking morpholino designed in our study has 76% sequence identity to the morpholino used in Wager et.al. The splice-blocking morpholino i2e3 has 88% sequence identity to the one used in Wager et al. As shown in the manuscript, we could not observe similar phenotypes neither with any of our tested morpholinos nor in maternal zygotic *cln3* mutants. From the Wager study, it therefore looks like MOs can exert off-target effects leading to phenotypes reminiscent of or resulting from lysosomal dysfunction. They can thus potentially be used to induce models for lipofuscinosis, but should not be linked directly to *cln3* deficiency.

How can the authors be sure the strains are true knockouts without using long read RNA or alternative primers in different regions, given that they did not use antibodies to test the presence or absence of CLN3 protein? Can they sequence check the *Cln3* locus? I strongly suggest using alternative names in case these turn out not to be true knockouts with absolutely no *Cln3* produced.

To confirm the mutations on the cDNA level, we designed primers binding to the ATG and stop codons of the Ensembl canonical *cln3* sequence (ENSART0000055170.4). We extracted RNA at 5 dpf and only one transcript in each line could be amplified (see agarose gel analysis of the amplicons in File S2 of the revised manuscript, also copied for your convenience below here) and those transcripts were fully sequenced. The MUT 1 allele encodes an early stop codon and the MUT2 allele contains an in-frame mutation, as described in our manuscript (page 8). We have also now included the full sequence data as supplemental information of the revised manuscript (File S2); the most relevant sequence information is shown in main Figure 2. Experimental validation of the presence or absence of a protein traditionally relies on Western blotting; but this requires a target-specific antibody, which is lacking for a majority of zebrafish proteins, including *Cln3*. We were also unable to detect *Cln3* protein in membrane extracts of 5 dpf WT larvae and of adult WT zebrafish brains using LC-MS/MS, most likely because of its low abundance and high hydrophobicity; so we could not use this methodology either to check for absence of *Cln3* protein in MUT1. Repeating such assays in lysosomal fractions may be a promising avenue. Our strongest argument in favor of a profound loss of *Cln3* function in our MUT1 and MUT2 lines is the high accumulation of GPDs in both of these lines, reaching more than 30-fold increases in the case of GPI (see Figure 6). However, we agree with the reviewer that we cannot exclude the presence of truncated forms of *Cln3* that retain some residual function, maybe not directly involved in GPD metabolism. MUT2 contains an in-frame deletion predicted to result in expression of a mutant protein lacking only exon 4, whereas it seems that MUT1 mainly expresses a much shorter truncated version of *Cln3*. The similar levels of GPD accumulation found in both lines were even surprising to us, as we would have expected a more moderate loss of function for MUT2. This indicates a quite important role of the first luminal loop, encoded by exon 4 of *cln3*, for maintaining normal GPD levels. Still, in the light of previous studies demonstrating residual function in maintaining normal lysosomes for shorter forms of *Cln3* (notably in Kitzmüller et al, 2008, results that we mention in the discussion of the paper), we further edited our text underlining that MUT1 is only a putative KO line. In addition, there may be transcripts with alternative start and/or stop codons that we could not have amplified with our primers and we agree that different types of analyses, such as long read RNAseq, would be needed to fully study this question, as is also now indicated in the revised manuscript (page 7, line 210-215).

File S2: Sanger sequencing of transcripts detected in zebrafish CRISPR mutants

Based on the canonical *cln3* transcript (ENSDART00000055170.4) described in Ensembl, a primer pair was designed to amplify the entire coding sequence following the Gateway guidelines. Total RNA was extracted from 5 dpf WT, MUT1, and MUT2 larvae and cDNA was synthesized. The PCR reaction was analyzed by agarose gel electrophoresis (see below). Amplicons were cloned into pDONR221 and recombinant plasmid was sequenced using M13 primers (see below). Predicted transcript length: WT, 1342 bp; MUT1, 1440 bp; MUT2, 1259 bp.

Agarose gel (2%) showing *cln3* amplicons amplified from WT, MUT1 and MUT2 cDNA.

For both e2i2-MO and i2e3-MO, there is a disruption within the primer targeting sequence and a shift is observed. However, the sequence of these PCR products are not shown to assess if the predicted Stop codon is actually introduced.

We confirmed the introduction of stop codons in the shortened transcripts detected in our morphants by Sanger sequencing. This information is now included in the File S1 joined to the revised manuscript.

Was exon skipping considered which is highly likely and could produce partially function *cln3* in all mutants including the CRISPRs? As in Kitzmuller et al and other CLN3 papers. Have theach

Yes, it was considered, and we understand the concerns of this reviewer. As explained in detail above, in response to the 'true knockout' comment:

- In the MUT1 line, we detect only one major transcript using primers that should amplify the entire CDS of the canonical *cln3* transcript listed in ENSEMBL; this transcript contains a premature stop codon leading to a very short, truncated form of *Cln3* (122 aa, including 33 non-native aa).
- In the MUT2 line, we expect production of a protein just lacking exon 4, as clearly stated in the manuscript. Nevertheless, even that mutant accumulates high levels of GPDs, to the same extent than MUT1, as we also now show in the revised manuscript (Figure 6).
- We used cautious language, not excluding the possibility of residual function.
- We designated our lines with the MUT abbreviation and avoided the KO abbreviation.

The authors made *slc45a2* mutants in the same fish that they were making *cln3* mutants. I am concerned that there is no mention of ensuring that the *cln3* mutant lines did not contain the *slc45a2*

mutation (even as heterozygotes). If it did, this would hamper interpretation of cln3 mutant phenotypes.

The same point was made in a comment above starting with "The chimeric PO fish have pigment changes...". Please refer to our answer above.

The authors have not followed ARRIVE guidelines as they do not state how many animals were used per experiment.

In the revised manuscript, we made sure that the number of animals used is indicated for all the presented experiments, in the corresponding figure legends.

p.7 Correct reference to 1.02kb - it is 1 kb deletion and there are at least two transcripts for this disease allele Kitzmuller et al 2008. Did you look for other transcripts arising from exon skipping??

We corrected 1.02kb to 1kb throughout. Concerning the exon skipping, please refer to our answers to the two previous comments raising this question.

P12. There is no reference for reduced CTSD activity in CLN3 Batten disease, which should be added, or instead as CLN10 disease is caused by mutations in CTSD then this is good reason to look.

We thank the reviewer for this suggestion, we revised the text in that results section (page 13, line 355-359) and included relevant references.

They should provide the transition tables for the targeted metabolite assays (GPDs most importantly) as they have only done this for the carnitines and AA assays. This is quite important for other groups to replicate/compare future findings.

In this study, targeted amino acid and GPD measurements were performed using high-resolution mass spectrometry. We did not use single or multiple reaction monitoring methods and therefore we did not need to define any mass transitions for our target metabolites. We have used the mass peak of the protonated (ESI+) or deprotonated (ESI-) molecules to quantify the specific, target compounds. As explained in the methods section of the manuscript, the identity of all targets was confirmed by accurate mass, MS/MS experiments, and standard addition experiments, when authentic standards were available. Briefly, annotation of the compounds of interest was achieved by isolating the precursor ion and fragmenting it into the product ions. The resulting product ion spectrum was used for library matching and qualitative MS interpretation. The procedural error was corrected by using the response ratio of the integrated peak area of the target compound and the integrated peak area of the dedicated internal standard. Again, this alternative (HRMS-based) method for targeted measurements does not involve "transitions" (as for Triple Quad-based SRM or MRM methods).

They should clarify the internal standards they have used for the targeted analysis - it is unclear what was used for the GPDs (IS has been used before).

For the targeted GPD measurements, deuterated citrulline was used as an internal standard; this is now indicated in the methods section (page 31, line 866-867). To confirm the chemical identity of the GPD peaks, external GPD standards were used. The ones that were not commercially available were

synthesized in-house based on a previously published method, as described in the manuscript (page 14, line 373-376; page 30, line 847-848)

Discussion:

FigS1A- add a comment that the alignment is agreement with Wager et al, and extends the analysis to further species.

We implemented the reviewer's comment in the results section of the revised manuscript (page 6, line 142-143).

Discuss RNA expression in FigS1B in relation to in situ hybridisation in Wager et al.

We implemented the reviewer's comment in the results section of the revised manuscript (page 6, line 150).

If the different background effect is not responsible for the difference in the Wager et al morphant studies, what could the reason be? Is it worth adding a sentence to say this is unexplained as it appears that the majority of transcripts are affected? Could the amount of reduction of normal transcripts in relation to the heterozygous state (which is phenotypically normal) be discussed, and compared to previous papers estimating how much CLN3 protein or transcript is required for normal development and absence of phenotypes.

We expanded our comparison of the morphant experiments in previous work and in this study in the discussion section (page 17, line 454-465), also in response to the first comment of reviewer #1, who recommended discussing the observed discrepancy in stronger terms. We want to highlight again that we used in our study a translation-blocking morpholino which failed to reproduce the lethal phenotype observed in the Wager study. Our stable mutants show metabolic phenotypes (GPD accumulation and BMP depletion) which can now be considered as conserved signatures of CLN3 deficiency. Taken together, these observations suggest off-target effects with the MOs used in the Wager study as a most plausible explanation for the discrepancy with our results, until the contrary can be demonstrated.

p.16 'tissular'??

We revised the sentence to omit this term (page 18, line 504 of the revised manuscript).

Figures:

2E Correct reference to 1.02kb - it is 1 kb deletion and there are at least two transcripts for this disease allele Kitzmuller et al 2008

We clarified in the legend of Figure 2 that the main transcript derived from the 1 kb deletion allele is presented in this figure.

2F The first fish is smaller than the others. Is this picture at the same magnification?

Yes, it was. The question concerning this size difference was addressed and clarified in response to an earlier comment on this point by this reviewer and the figure was revised.

S2 shows 11 hydrophobic regions but Fig 2E shows only six. Reconcile?

We agree that this is confusing in our original manuscript. We have adapted Figure 2 to our hydrophobicity prediction analysis presented in Fig S3 (11 hydrophobic regions).

S1A show exact species in legend and ideally also the figure. Clarify if point mutations (ie single nucleotide changes) or missense mutations affecting a single amino acid. Indicate 1 kb deletion. There are other potential posttranslational sites, so either show all or none

Exact species names and protein identifiers of the sequences shown are now indicated in the figure legend. We have changed the designation of the aa residues highlighted in red to 'known missense mutation in CLN3 Batten disease'. Amino acids preserved in the major transcript of the 1 kb deletion allele of human CLN3 are now underlined in the figure (red line). Indeed, there are other putative posttranslational modifications that are described in the literature such as phosphorylation, O-glycosylation sites at T80 and T256, and myristoylation (Mirza et al., 2019). However, we decided to represent only the most well documented posttranslational sites described in the literature (i.e. the N-glycosylation and farnesylation sites) (Calcagni et al., 2023; Jarvela et al., 1998; Mirza et al., 2019). We specify this now in the revised legend of the figure. We believe that this representation can also help the reader to understand better the molecular consequences of the MUT2 mutation, which is missing critical PTM sites and, based on the existing literature, may therefore not be correctly delivered into the lysosomes. The other posttranslational modifications are less well documented in the literature for the human CLN3 protein and fewer studies in other models.

Figure S2. Conservation and expression of the zebrafish *cln3* gene. (A) Multiple sequence alignment of CLN3 protein sequences shows high conservation across various species (Yeast, *Saccharomyces cerevisiae* sequence NP_012476.1; Zebrafish, *Danio rerio* sequence NP_001007307; Human, *Homo sapiens* sequence NP_001273038; Mouse, *Mus musculus* sequence NP_001139783). Gaps are indicated with dashes and different shades of blue were used to represent conservation level: dark blue, blue, and light blue for 100%, 75% and 50% conservation in the selected sequences, respectively. Residues in the human sequence known to harbor missense mutations in CLN3 Batten disease are highlighted in red. The red line indicates the 153 amino acids preserved from the wild-type protein in the major transcript expressed from the 1 kb deletion allele of human CLN3. Posttranslational modification sites are highlighted in green and violet. The main experimentally validated posttranslational modifications are indicated (Mirza et al., 2019, Calcagni et al., 2023, Jarvela et al., 1998). For the zebrafish Cln3 sequence, residues 1-122 may be preserved in a putative truncated expression product in the MUT1 line and residues 62-90 are missing in the predicted variant expressed by the MUT2 line described in this study

It would be better to report normalised abundance/ 40 larvae in the graph axis instead of mentioning in the legend.

The y-axis annotation suggested by the Reviewer could be misleading. We pooled 40 larvae (per replicate) to extract the metabolites, but we did not normalize the metabolites by the number of larvae. Instead, the peak intensities of the target metabolites were normalized to the summed peak area of all compounds, as described in the Methods section. We refer to this Methods section in the legend to Fig. 5 now for enhanced clarity.

SSA a 3-D PCA on a 2D page needs the 3rd vector direction indicated. Perhaps present as a biplot as most of the separation is in the PC1 (25.7%).

We agree with the reviewer that PC1 explains most of the variance between the two groups. We used the PCA primarily as a visualization tool to probe if we can observe a separation between the two groups based on our untargeted metabolomics data. To note, the variance of PC2 and PC3 are indicated on the plot (14.9% and 10.3%, respectively). We have improved the orientation of the figure and the annotated axes can now be distinguished better. We agree that a biplot could be another visualization tool, we kept the current figure since it seems to be fit for our purpose as well.

Methods:

It would be useful to indicate introns versus UTR and coding sequences in the MO sequences e.g. by using upper and lower case.

The suggestion was integrated in the revised manuscript and UTR and intron sequences are indicated with lower case in the MO sequences (page 23, line 632-634).

What strain of fish was injected to generate P0 fish? Were CRISPR targets sequenced in the strain that were injected? Were F1 and F2s checked for lack of slc45a2 mutations? If so, how?

As indicated in the Methods section 'Zebrafish lines and handling', we performed all our experiments in the AB background. Yes, we sequenced the Crispr target sites in the MUT1 and MUT2 lines and the results are shown in main Figure 2C. For demonstrating lack of slc45a2 mutations, please refer to our answer to the same question above.

It would be useful to state how the mutants and heterozygotes were generated e.g. by incrossing heterozygotes, so they were siblings raised together, or were some homozygous mutants generated by incrossing homozygous adults, or homozygous x heterozygous adults. Were all the mutants zygotic, or were some maternal zygotic?

We have included all the details about mutant line generation in the revised manuscript (page 22, line 622-627; Fig S2A; and File S4). Heterozygous F1 fish (obtained by crossing P0 fish with wild-type fish) were incrossed to obtain an F2 generation which was genotyped and grouped accordingly (WT, heterozygous, and homozygous mutant). Homozygous mutant F2 animals were incrossed to generate F3 larvae for the experiments or raised for the creation of a stable F3-F4 homozygous mutant line. For the behavioral assays, to have the most similar genetic background possible in our experiments, we outcrossed F2 homozygous mutant fish with sibling F2 WT fish to generate F3 heterozygous larvae as the control reference in our experiments. We maintain the MUT lines in homozygous state and every fifth generation we outcross animals with WT AB animals to avoid inbreeding defects. For each mutant generation, respective WT siblings are raised in parallel.

Reviewer #3 (Comments to the Authors (Required)):

Zebrafish have been successfully used to model a range of neurological and metabolic perturbations in different lysosomal storage diseases (Zhang, et al 2020). Heins-Marroquin and colleagues aimed at developing and characterizing a zebrafish model of Cln3 disease at the early developmental stages of life. Heins- Marroquin et al mutant models exhibit milder behavioral abnormalities than another published line (Wager, et al 2016). The mutant lines also recapitulate the major metabolic changes that

were previously observed in human patients and some other mammalian *cln3*-deficient systems mainly, the abnormal levels of GPDs and BMPs.

Major comments to improve the manuscript:

I don't think the study is particularly novel and no notable technological advance acquired as much as I can see. The work has been done on the whole larvae level and no work at the subcellular level, though *Cln3* is a lysosomal protein.

In this study, we did not aim to achieve notable technological advances. The main objectives and achievements of our work are described in the last paragraph of the introduction. At this stage, the GPD biomarker findings are not novel anymore, but given that this is still a very recent and potentially breakthrough development in the *CLN3* research community, it will be of high relevance to the latter that we could demonstrate here that these compounds can be detected at very early stages also in an excellent developmental model (zebrafish) as well as in human cerebral organoids with *CLN3* deficiency.

1. The authors could visualize (image) the lysosomes (morphology, size, count, volume) in WT or Het versus MUT

We agree with the reviewer that this work is highly relevant in this model. Our focus for this initial study was on the generation of stable mutants and behavioral as well as metabolomic studies in larval stages, relying on main domains of expertise of the author team. We only started to look into lysosomal features, and we have found interesting but preliminary disease-related phenotypes that are shown in Figure 2 below: mild accumulation of SCMA levels in *cln3* adult zebrafish whole brain (Fig. 2A,B), increased lysosomal size of macrophages (lysotracker staining) in tailfin injury assay or increased immune cell migration to the site of injury as stronger response to inflammation (Fig. 2C,D). The latter phenotype seems very promising but currently we cannot distinguish if we have a higher agglomeration of macrophages or bigger lysosomes as we can only visualize the lysosomes with the lysotracker. The use of transgenic zebrafish reporter lines with fluorescently labeled immune cells is needed to progress further in this work. For the creation of transgenic mutant lines, we are outcrossing *cln3* homozygous zebrafish with the *mpeg:GFP-CAAX* reporter line, which will be followed by incrossing to obtain a homozygous mutant reporter fish line. The generation of the line is estimated to take 6-8 months. In addition, we show in the manuscript that preliminary electron microscopy analyses in 5 dpf MUT1 larvae did not reveal the fingerprint like inclusions that can be seen in other models of this type of lysosomal storage disease. These observations indicate that lysosomal changes are not evident at the early larval stage, but we agree that it will be essential to follow up on this preliminary work with more systematic studies at this and later stages of development as well as in adulthood.

[Figure removed by editorial staff per authors' request]

2. Besides the whole level, the author could use lysosome enrichment kit for lysosomal purification from tissues, and subsequent metabolic analyses. In particular, the major changing metabolites (GPDs) are trapped within the lysosome. Lysosomes represent a small fraction of the whole cell, therefore, the lysosomal changes will be diluted at the whole cell level. The enrichment for lysosomes will improve the results and add more value for using this zebrafish model to monitor the disease. This part has been discussed in Laqtom et al, 2022.

We fully agree with the reviewer that adapting the LysoIP method for rapid isolation of lysosomes from zebrafish larvae or tissues is an extremely promising perspective in the context of CLN3 and other lysosomal storage diseases. However, generating and validating a LysoTag zebrafish and then exploiting it in the context of CLN3 deficiency analogously to what has been done for mouse in Laqtom et al (2022) will realistically take more than a year of work and goes largely beyond the scope of the manuscript under review here. In a way, one of the beauties of this study and our models is that even without doing lysosomal enrichment, we could readily detect in extracts of whole 5-dpf larvae, very significantly increased levels of GPDs and decreased levels of BMPs, showing even more strikingly how important these changes are given the small volume that lysosomes represent compared to the whole cell. A LysoTag zebrafish will be the next step in drilling deeper into more subtle lysosomal changes that cannot be revealed at the 'whole cell' level, but it is not needed to convincingly make the points that we want to publish in this first study.

3. The authors generated a comprehensive list of dysregulated polar and non-polar metabolites, I would encourage the authors to follow up on novel results and reproduce them in their organoid model system.

We fully agree with the reviewer. This work is ongoing but the results are not ready for publication at this stage.

4. I think this manuscript is long and can be written concisely.

We did an effort to shorten the manuscript in several parts:

Page 13, paragraph 2 was shortened. We fused two results sections together ('Metabolic perturbations in *cln3* mutant larvae detected by untargeted metabolomics' and 'Validation of metabolic disruption and assessment of lysosomal proteolytic activity in *MUT1* zebrafish larvae'), not describing the second untargeted metabolomics experiment anymore as a validation experiment, but rather as an independent replicate confirming the results of the first one. This responds to a comment as well of Reviewer 1 and allowed to further shorten the manuscript.

Page 15, paragraph 2. Explanation for the confirmation of the identity of the statistically significant metabolites was simplified.

Page 20, paragraph 1 was removed as some of these metabolite changes could not be validated and the text was speculative.

However, in response to other reviewers' comments we also needed to add new text in other parts so that overall, the manuscript has not been shortened significantly in the end.

Minor comments:

Sup Fig. 2B, are both reference genes stably expressed in all times of developmental stages? Show statistical significance in the figure.

Assuming that the reviewer refers to Sup Fig. 1B:

- From the literature, the 2 reference genes that we chose should be stably expressed in all times of developmental stages (Xu et al., 2016, McCurley and Callard, 2008).
- Statistical significance is now indicated in the figure of the revised manuscript.

Fig S2. (B) Zebrafish *cln3* expression levels at different developmental stages are represented relative to the expression level at 6 hpf. (C-D) Zebrafish *cln3* expression levels in the indicated organs are shown relative to the expression levels in the eye. RNA was extracted from the organs of 2-year-old male and female zebrafish for qPCR analysis. All expression levels were normalized to either the *rpl13α* (dark blue) or the *ef1α* (dotted light blue) reference genes, which were reported to be stably expressed during development and across tissues (McCurley and Callard, 2008, Xu et al., 2016). Data shown in panel B are technical replicates of a pool of 30 larvae. Data shown in panels (C) and (D) are means \pm SDs from three technical replicates of a pool of 30 larvae (B) or from biological replicates of 3 adult fish (C, D). Statistical significance was estimated using the two-way ANOVA test (*, $p \leq 0.05$; **, $p \leq 0.01$; ***, $p \leq 0.001$; ****, $p < 0.0001$). hpf, hours post-fertilization.

In Discussion:

In the *Cln3* knockout mouse model, tissular GPD accumulation became evident only in samples enriched in lysosomes (lysoIP approach) and was shown for approximately 7-months old animals (Laqtom et al. 2022).

I think 'only' is inaccurate here. By looking at their Fig.3, there were statistically increased levels of GPDs in the whole tissue and cell, but it becomes much more evident in lysosomal-enriched samples.

We agree with the reviewer's comment and have revised the sentence accordingly (page 19, line 504-505).

References

- CALCAGNI, A., STAIANO, L., ZAMPELLI, N., MINOPOLI, N., HERZ, N. J., DI TULLIO, G., HUYNH, T., MONFREGOLA, J., ESPOSITO, A., CIRILLO, C., BAJIC, A., ZAHABIYON, M., CURNOCK, R., POLISHCHUK, E., PARKITNY, L., MEDINA, D. L., PASTORE, N., CULLEN, P. J., PARENTI, G., DE MATTEIS, M. A., GRUMATI, P. & BALLABIO, A. 2023. Loss of the batten disease protein CLN3 leads to mis-trafficking of M6PR and defective autophagic-lysosomal reformation. *Nat Commun*, 14, 3911.
- EISEN, J. S. & SMITH, J. C. 2008. Controlling morpholino experiments: don't stop making antisense. *Development*, 135, 1735-43.
- HEINS-MARROQUIN, U., JUNG, P. P., CORDERO-MALDONADO, M. L., CRAWFORD, A. D. & LINSTER, C. L. 2019. Phenotypic assays in yeast and zebrafish reveal drugs that rescue ATP13A2 deficiency. *Brain Commun*, 1, fcz019.

- IRION, U., KRAUSS, J. & NUSSLEIN-VOLHARD, C. 2014. Precise and efficient genome editing in zebrafish using the CRISPR/Cas9 system. *Development*, 141, 4827-30.
- JAO, L. E., WENTE, S. R. & CHEN, W. 2013. Efficient multiplex biallelic zebrafish genome editing using a CRISPR nuclease system. *Proc Natl Acad Sci U S A*, 110, 13904-9.
- JARVELA, I., SAINIO, M., RANTAMAKI, T., OLKKONEN, V. M., CARPEN, O., PELTONEN, L. & JALANKO, A. 1998. Biosynthesis and intracellular targeting of the CLN3 protein defective in Batten disease. *Hum Mol Genet*, 7, 85-90.
- MCCURLEY, A. T. & CALLARD, G. V. 2008. Characterization of housekeeping genes in zebrafish: male-female differences and effects of tissue type, developmental stage and chemical treatment. *BMC Mol Biol*, 9, 102.
- MIRZA, M., VAINSHTEIN, A., DIRONZA, A., CHANDRACHUD, U., HASLETT, L. J., PALMIERI, M., STORCH, S., GROH, J., DOBZINSKI, N., NAPOLITANO, G., SCHMIDTKE, C. & KERKOVICH, D. M. 2019. The CLN3 gene and protein: What we know. *Mol Genet Genomic Med*, 7, e859.
- MORENO-MATEOS, M. A., VEJNAR, C. E., BEAUDOIN, J. D., FERNANDEZ, J. P., MIS, E. K., KHOKHA, M. K. & GIRALDEZ, A. J. 2015. CRISPRscan: designing highly efficient sgRNAs for CRISPR-Cas9 targeting in vivo. *Nat Methods*, 12, 982-8.
- VICENCIO, J., SANCHEZ-BOLANOS, C., MORENO-SANCHEZ, I., BRENA, D., VEJNAR, C. E., KUKHTAR, D., RUIZ-LOPEZ, M., COTS-PONJOAN, M., RUBIO, A., MELERO, N. R., CRESPO-CUADRADO, J., CAROLIS, C., PEREZ-PULIDO, A. J., GIRALDEZ, A. J., KLEINSTIVER, B. P., CERON, J. & MORENO-MATEOS, M. A. 2022. Genome editing in animals with minimal PAM CRISPR-Cas9 enzymes. *Nat Commun*, 13, 2601.
- XU, H., LI, C., ZENG, Q., AGRAWAL, I., ZHU, X. & GONG, Z. 2016. Genome-wide identification of suitable zebrafish *Danio rerio* reference genes for normalization of gene expression data by RT-qPCR. *J Fish Biol*, 88, 2095-110.

December 11, 2023

RE: Life Science Alliance Manuscript #LSA-2023-02057-TR

Dr. Carole Liette Linster
University of Luxembourg
Luxembourg Centre for Systems Biomedicine
7, Ave des Hauts-Fourneaux
Esch-sur-Alzette 4362
Luxembourg

Dear Dr. Linster,

Thank you for submitting your revised manuscript entitled "CLN3 deficiency leads to neurological and metabolic perturbations during early development". We would be happy to publish your paper in Life Science Alliance pending final revisions necessary to meet our formatting guidelines.

-please use the [10 author names, et al.] format in your references (i.e. limit the author names to the first 10)
please update your callouts for the Supplementary Figures in the manuscript Fig s6 a,b,c; Fig s7 a,b

A. FINAL FILES:

B. MANUSCRIPT ORGANIZATION AND FORMATTING:

****It is Life Science Alliance policy that if requested, original data images must be made available to the editors. Failure to provide**

original images upon request will result in unavoidable delays in publication. Please ensure that you have access to all original data images prior to final submission.**

The license to publish form must be signed before your manuscript can be sent to production. A link to the electronic license to publish form will be available to the corresponding author only. Please take a moment to check your funder requirements.

Sincerely,

Reviewer #1 (Comments to the Authors (Required)):

The extensively revised version of the manuscript largely improves the manuscript. It addresses my concerns and I recommend publication.

Reviewer #2 (Comments to the Authors (Required)):

This revised paper describes the generation of morphant models and CRISPR-generated CLN3 lines of Zebrafish which unexpectedly showed no obvious gross phenotypes. The most striking finding was that in-depth metabolomics and lipidomics analyses revealed significant accumulation of several glycerophosphodiester (GPDs), especially GPI and a global decrease of bis(monoacylglycero)phosphate (BMP) species. This supports their utility as potential biomarkers, and is consistent with recent publications in the field.

The revised paper is well written and contains an even greater wealth of well presented data throughout.

The authors have done an admirable job of addressing (or explaining why it is not appropriate) the reviewers' comments. This has further improved an excellent paper.
In my opinion, there are no further issues to address.

Reviewer #3 (Comments to the Authors (Required)):

The authors have been responsive to comments. I support publication and congratulate them on the scientific contribution to the field!

December 13, 2023

RE: Life Science Alliance Manuscript #LSA-2023-02057-TRR

Dr. Carole Liette Linster
University of Luxembourg
Luxembourg Centre for Systems Biomedicine
6, Ave du Swing
Belvaux 4367
Luxembourg

Dear Dr. Linster,

Thank you for submitting your Research Article entitled "CLN3 deficiency leads to neurological and metabolic perturbations during early development". It is a pleasure to let you know that your manuscript is now accepted for publication in Life Science Alliance. Congratulations on this interesting work.

DISTRIBUTION OF MATERIALS:

Again, congratulations on a very nice paper. I hope you found the review process to be constructive and are pleased with how the manuscript was handled editorially. We look forward to future exciting submissions from your lab.

Sincerely,
